# An intranuclear bacterial parasite of deep-sea mussels expresses apoptosis inhibitors acquired from its host

Miguel Ángel González Porras [1], Adrien Assié [1,2], Målin Tietjen[1], Marlene Violette [3], Manuel Kleiner[3], Harald Gruber-Vodicka[1,4], Nicole Dubilier [1] ✉ & Nikolaus Leisch [1,5] ✉

A limited number of bacteria are able to colonize the nuclei of eukaryotes. 'Candidatus Endonucleobacter' infects the nuclei of deep-sea mussels, where it replicates to ≥80,000 bacteria per nucleus and causes nuclei to swell to 50 times their original size. How these parasites are able to replicate and avoid apoptosis is not known. Dual RNA-sequencing transcriptomes of infected nuclei isolated using laser-capture microdissection revealed that 'Candidatus Endonucleobacter' does not obtain most of its nutrition from nuclear DNA or RNA. Instead, 'Candidatus Endonucleobacter' upregulates genes for importing and digesting sugars, lipids, amino acids and possibly mucin from its host. It likely prevents apoptosis of host cells by upregulating 7–13 inhibitors of apoptosis, proteins not previously seen in bacteria. Comparative phylogenetic analyses revealed that 'Ca. Endonucleobacter' acquired inhibitors of apoptosis through horizontal gene transfer from their hosts. Horizontal gene transfer from eukaryotes to bacteria is assumed to be rare, but may be more common than currently recognized.

Most metazoans are intimately associated with bacteria[1], and some of these live inside eukaryotic cells, but only very rarely inside eukaryotic organelles[2–4]. Marine animals are often associated with a family of Gammaproteobacteria fittingly named Endozoicomonadaceae. Most Endozoicomonadaceae are extracellular, and only a few *Endozoicomonas* species and their close relatives live inside their host's cells[5]. First isolated from a sea slug only 17 years ago[6], *Endozoicomonas* and other Endozoicomonadaceae have been revealed by culture-independent sequencing approaches to be ubiquitous and common inhabitants of a wide diversity of marine animals, from sponges and corals to fish[7]. Their role for their hosts has often been inferred but rarely proven, and is described as ranging from parasitic and commensalistic to beneficial[7,8]. All cultured Endozoicomonadaceae are aerobic, or facultatively anaerobic, heterotrophs that were isolated from marine hosts[9].

A single clade of Endozoicomonadaceae, 'Candidatus Endonucleobacter', lives inside its host's nuclei. These bacteria infect the nuclei of deep-sea bathymodioline mussels from hydrothermal vents and cold seeps around the world[10]. The 'Ca. Endonucleobacter' infection cycle begins with a single bacterium that invades the nucleus and then grows by elongating and dividing. In the final stages of infection, the elongated cells undergo septated division and replicate to as many as 80,000 cells, causing the mussel's nuclei to swell to as much as 50 times their original size. Eventually, the infected mussel cells burst, releasing 'Ca. Endonucleobacter' into the seawater[10].

Intranuclear bacteria have rarely been described in animals but are well known from protists[11–13]. In protists, the bacteria belong to other bacterial lineages than the gammaproteobacterial 'Ca. Endonucleobacter', such as the Rickettsiales, Holosporales and Verrucomicrobiota,

[1]Max Planck Institute for Marine Microbiology, Bremen, Germany. [2]Alkek Center for Metagenomics and Microbiome Research, Baylor College of Medicine, Houston, TX, USA. [3]Department of Plant and Microbial Biology, North Carolina State University, Raleigh, NC, USA. [4]Zoological Institute, Christian-Albrechts-University, Kiel, Germany. [5]European Molecular Biology Laboratory, Heidelberg, Germany. ✉e-mail: ndubilie@mpi-bremen.de; leisch@embl.de

and do not replicate to such high numbers within their host's nuclei as '*Ca.* Endonucleobacter'. To date, nothing is known about the molecular and cellular processes that intranuclear bacteria of animals use to infect and reproduce in their host. Key questions are how bacteria that infect animal nuclei are able to counter host immune responses, avoid the induction of host cell death through apoptosis and gain nutrition for their massive replication. It was hypothesized that '*Ca.* Endonucleobacter' digests nuclear chromatin, but this would quickly impair the cellular activity of the host cell, including its immune responses[10,12]. Moreover, the deformation of the host cytoskeleton that '*Ca.* Endonucleobacter' induces through the dramatic increase in nuclear volume would trigger apoptosis, a common response of metazoans to infection by parasites[14–17]. Chromatin digestion and induction of apoptosis would quickly lead to the death of infected cells, thus preventing '*Ca.* Endonucleobacter' from replicating to such high numbers.

To reveal the genetic adaptations that allow '*Ca.* Endonucleobacter' to thrive in its intranuclear niche, we assembled high-quality genomes of two '*Ca.* Endonucleobacter' species, specific to two bathymodioline host species, *Bathymodiolus puteoserpentis* from hydrothermal vents on the Mid-Atlantic Ridge and *Gigantidas childressi* from cold seeps in the Gulf of Mexico, and compared them to closely related Endozoicomonadaceae. To gain insights into the metabolism of '*Ca.* Endonucleobacter', we analysed the metatranscriptomes and metaproteomes of bulk gill tissues from *G. childressi*. Finally, to understand host–microorganism interactions during the infection cycle of '*Ca.* Endonucleobacter', we used laser-capture microdissection, coupled with ultra-low-input dual RNA-sequencing (RNA-seq), to generate transcriptomes of both the parasite and the host in early, middle and late infection stages (Fig. 1a and Supplementary Video 1).

## Results

### Two '*Ca.* Endonucleobacter' species with different infection patterns

We used fluorescence in situ hybridization (FISH) analyses, with probes specific to '*Ca.* Endonucleobacter', to analyse its distribution in *B. puteoserpentis* and *G. childressi*. These deep-sea mussels, like all other bathymodioline species investigated so far, house symbiotic sulfur- and/or methane-oxidizing bacteria in gill cells that provide them with nutrition[18]. Our FISH analyses of thousands of cells from at least ten mussel individuals collected over several decades revealed that in both mussel species, the parasite never infected cells with symbiotic bacteria (Extended Data Fig. 1). The inability of '*Ca.* Endonucleobacter' to infect host cells with symbionts is thus not only consistent across two mussel species from different genera that are geographically separated by thousands of kilometres (Supplementary Table 1), but also consistent across symbiont types, with *G. childressi* harbouring only methane-oxidizing symbionts and *B. puteoserpentis* harbouring both methane- and sulfur-oxidizing symbionts. The inability to infect symbiont-containing cells is also consistent over time, as *B. puteoserpentis* mussels collected 13 years before the *B. puteoserpentis* examined here had the same distribution[10]. One explanation for this exclusion pattern could be that the apical surfaces of symbiont-containing bacteriocytes differ from those of other epithelial cells in bathymodoline mussels.

Their bacteriocytes lack both the cilia and microvilli typical of epithelial cell surfaces[19]. Epithelial surface structures are often targeted by pathogens for entering eukaryotic cells, and their absence could hinder '*Ca.* Endonucleobacter' from infecting cells with symbionts.

In *G. childressi*, '*Ca.* Endonucleobacter' was always restricted to the outer ciliated edges of the gill (Extended Data Fig. 1a–c), while in *B. puteoserpentis*, the parasite was distributed evenly across gill tissues (Extended Data Fig. 1d–f). The confinement of '*Ca.* Endonucleobacter' to the outer edges of the gill in *G. childressi* was fortunate because it allowed us to gain samples from these non-model, deep-sea hosts that were greatly enriched in the parasite, thus providing enough DNA for long-read sequencing and enabling the dual RNA-seq approach of the infectious cycle described below (Fig. 1a).

Our analyses of high-quality draft metagenome-assembled genomes (Supplementary Table 2), assembled from both short- and long-read sequencing of *B. puteoserpentis* and *G. childressi* gill tissues, revealed that these two mussel species are infected by genetically distinct '*Ca.* Endonucleobacter' species, based on their average nucleotide identity of only 84.3%. We named the two '*Ca.* Endonucleobacter' species after the host species in which they occur, '*Candidatus* Endonucleobacter puteoserpentis' in *B. puteoserpentis* and '*Candidatus* Endonucleobacter childressi' in *G. childressi* (Supplementary Note 1). A comparative phylogenomic analysis of the two '*Ca.* Endonucleobacter' species and 42 publicly available genomes of close relatives placed both '*Ca.* Endonucleobacter' species in a monophyletic clade within the family Endozoicomonadaceae (class Gammaproteobacteria), with the genus *Endozoicomonas* as their closest relatives (Fig. 1b and Supplementary Table 2). '*Ca.* Endonucleobacter' genomes were smaller, had reduced guanine–cytosine (GC) contents and encoded considerably less amino acid synthesis pathways than *Endozoicomonas* species (Fig. 1b). A detailed, comprehensive analysis of genome reduction in '*Ca.* Endonucleobacter' is planned in a future study to predict its impact on metabolic pathways in these intranuclear pathogens.

### '*Ca.* Endonucleobacter' gains nutrition from its host

How does '*Ca.* Endonucleobacter' gain energy and nutrition within the nucleus for its massive replication from one to more than 80,000 cells? Our metabolic reconstruction of the genomes of the two '*Ca.* Endonucleobacter' species, as well as the transcriptomes and proteomes of '*Ca.* E. childressi', revealed that nuclear DNA, RNA and histones are unlikely to be their main source of nutrition (see below). Instead, these intranuclear parasites likely import and consume sugars, lipids and amino acids from their host (Fig. 2 and Supplementary Tables 3 and 4). The eukaryotic nuclear pore complexes allow the passage of small molecules (≤30–60 kDa) between the nucleus and the cytoplasm[20], providing '*Ca.* Endonucleobacter' with access to not only nuclear but also many cytoplasmic molecules.

'*Ca.* E. childressi' and '*Ca.* E. puteoserpentis' are predicted to share highly similar metabolic pathways. They are both chemoorganoheterotrophs that encoded genes involved in glycolysis, the pentose phosphate pathway, tricarboxylic acid (TCA) cycle and aerobic respiration with oxygen as the terminal electron acceptor (Fig. 2 and Supplementary Tables 3 and 4). Both parasites encoded lipid and sugar importers,

**Fig. 1 | '*Ca.* Endonucleobacter' infectious cycle and phylogenomic analysis.** A single '*Ca.* Endonucleobacter' infects the mussel nucleus (early infection stage), grows through elongation and division (mid-infection stage), and finally divides through septation of the elongated cells to as many as 80,000 cells (late infection stage). In the final infection stage, the nucleus is enlarged by as much as 50-fold in volume, the host cell bursts and the parasites are released to the environment. **a**, '*Ca.* Endonucleobacter' infectious cycle in the early, mid and late stages of infection, shown in the left, middle and right columns, respectively, of the top row (middle row, FISH images of '*Ca.* E. puteoserpentis'; bottom row, TEM images of '*Ca.* E. childressi'). FISH with specific probes shows the parasite (in yellow) inside mussel nuclei (DAPI-stained DNA in blue) and neighbouring symbiont-containing cells (indicated with dotted lines) with sulfur-oxidizing symbionts (in green) and methane-oxidizing symbionts (in pink) (sequences of all FISH probes are listed in Supplementary Table 6). **e**, '*Ca.* Endonucleobacter' cell; c, chromatin; ne, nuclear envelope. The results are representative of five independent experiments. Scale bars, 1 μm. **b**, Phylogenomic analysis using 172 conserved marker genes shared between the two '*Ca.* Endonucleobacter' genomes and those of 42 closely related Endozoicomonaceae. Genes were identified and aligned with the GToTree pipeline, the tree was calculated with IQ-TREE and branch support (1,000 replicates) was calculated with both SH-aLRT and UFBoot. Six *Oceanospirillum* genomes were used to root the tree. Scale bars indicate substitutions per site. Key genome characteristics are listed at the right. A full tree with all bootstrap values is shown in Supplementary Fig. 3.

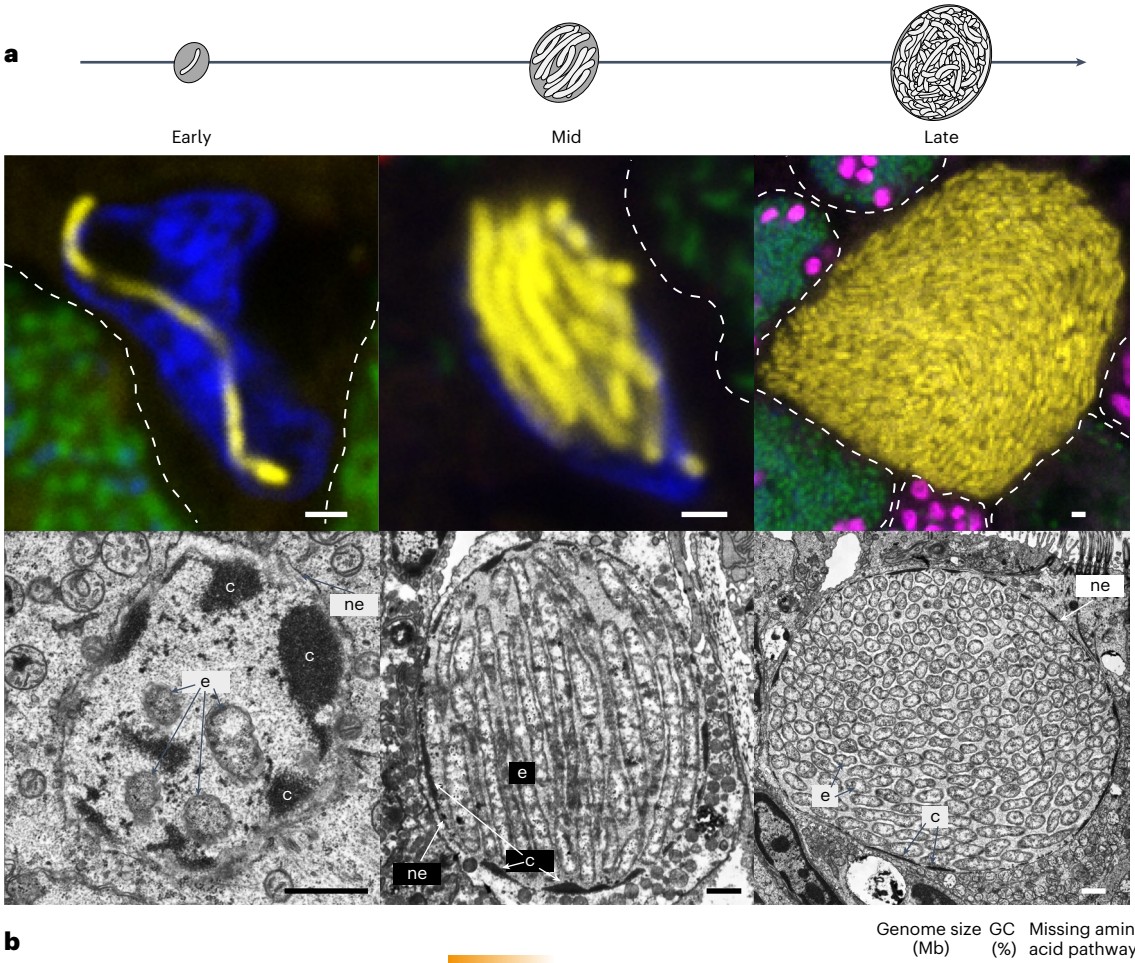

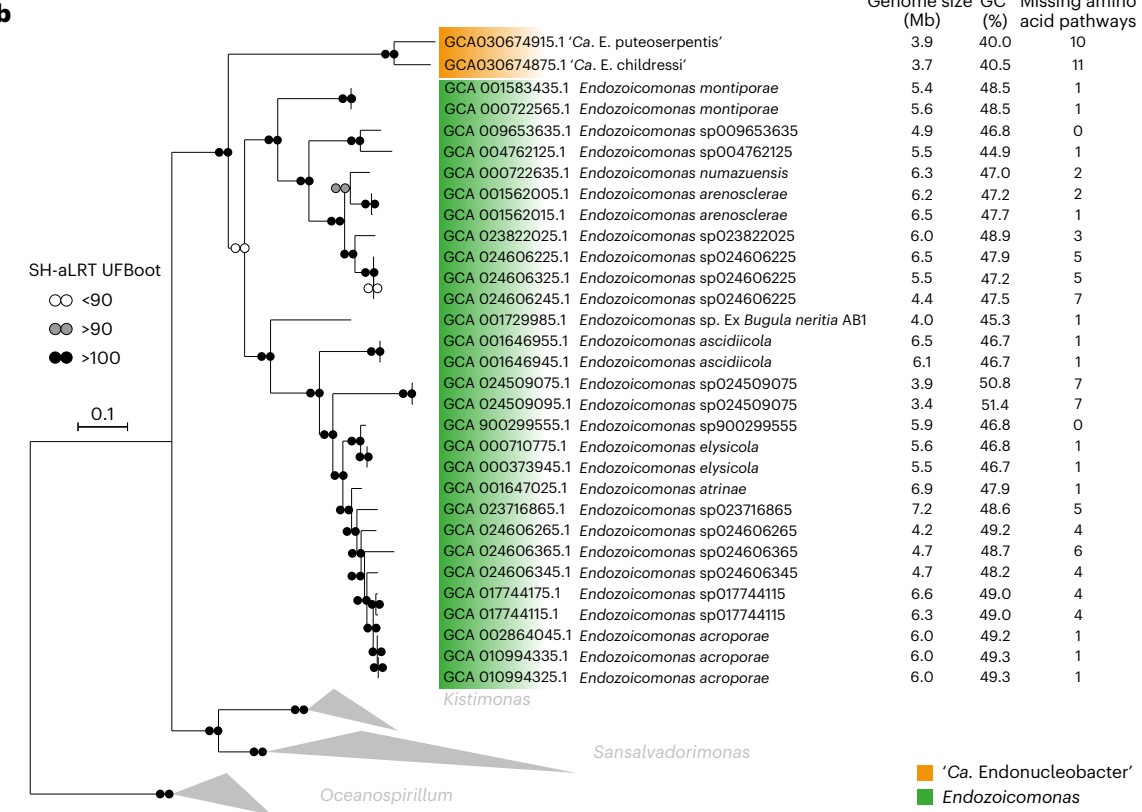

**b**

| | Genome size (Mb) | GC (%) | Missing amino acid pathways |
|---|---|---|---|
| GCA030674915.1 'Ca. E. puteoserpentis' | 3.9 | 40.0 | 10 |
| GCA030674875.1 'Ca. E. childressi' | 3.7 | 40.5 | 11 |
| GCA 001583435.1 Endozoicomonas montiporae | 5.4 | 48.5 | 1 |
| GCA 000722565.1 Endozoicomonas montiporae | 5.6 | 48.5 | 1 |
| GCA 009653635.1 Endozoicomonas sp009653635 | 4.9 | 46.8 | 0 |
| GCA 004762125.1 Endozoicomonas sp004762125 | 5.5 | 44.9 | 1 |
| GCA 000722635.1 Endozoicomonas numazuensis | 6.3 | 47.0 | 2 |
| GCA 001562005.1 Endozoicomonas arenosclerae | 6.2 | 47.2 | 2 |
| GCA 001562015.1 Endozoicomonas arenosclerae | 6.5 | 47.7 | 1 |
| GCA 023822025.1 Endozoicomonas sp023822025 | 6.0 | 48.9 | 3 |
| GCA 024606225.1 Endozoicomonas sp024606225 | 6.5 | 47.9 | 5 |
| GCA 024606325.1 Endozoicomonas sp024606225 | 5.5 | 47.2 | 5 |
| GCA 024606245.1 Endozoicomonas sp024606225 | 4.4 | 47.5 | 7 |
| GCA 001729985.1 Endozoicomonas sp. Ex Bugula neritia AB1 | 4.0 | 45.3 | 1 |
| GCA 001646955.1 Endozoicomonas ascidiicola | 6.5 | 46.7 | 1 |
| GCA 001646945.1 Endozoicomonas ascidiicola | 6.1 | 46.7 | 1 |
| GCA 024509075.1 Endozoicomonas sp024509075 | 3.9 | 50.8 | 7 |
| GCA 024509095.1 Endozoicomonas sp024509075 | 3.4 | 51.4 | 7 |
| GCA 900299555.1 Endozoicomonas sp900299555 | 5.9 | 46.8 | 0 |
| GCA 000710775.1 Endozoicomonas elysicola | 5.6 | 46.8 | 1 |
| GCA 000373945.1 Endozoicomonas elysicola | 5.5 | 46.7 | 1 |
| GCA 001647025.1 Endozoicomonas atrinae | 6.9 | 47.9 | 1 |
| GCA 023716865.1 Endozoicomonas sp023716865 | 7.2 | 48.6 | 5 |
| GCA 024606265.1 Endozoicomonas sp024606265 | 4.2 | 49.2 | 4 |
| GCA 024606365.1 Endozoicomonas sp024606365 | 4.7 | 48.7 | 6 |
| GCA 024606345.1 Endozoicomonas sp024606345 | 4.7 | 48.2 | 4 |
| GCA 017744175.1 Endozoicomonas sp017744115 | 6.6 | 49.0 | 4 |
| GCA 017744115.1 Endozoicomonas sp017744115 | 6.3 | 49.0 | 4 |
| GCA 002864045.1 Endozoicomonas acroporae | 6.0 | 49.2 | 1 |
| GCA 010994335.1 Endozoicomonas acroporae | 6.0 | 49.3 | 1 |
| GCA 010994325.1 Endozoicomonas acroporae | 6.0 | 49.3 | 1 |

SH-aLRT UFBoot
○○ <90
◐◐ >90
●● >100

0.1

*Kistimonas*

*Sansalvadorimonas*

*Oceanospirillum*

■ 'Ca. Endonucleobacter'
■ *Endozoicomonas*

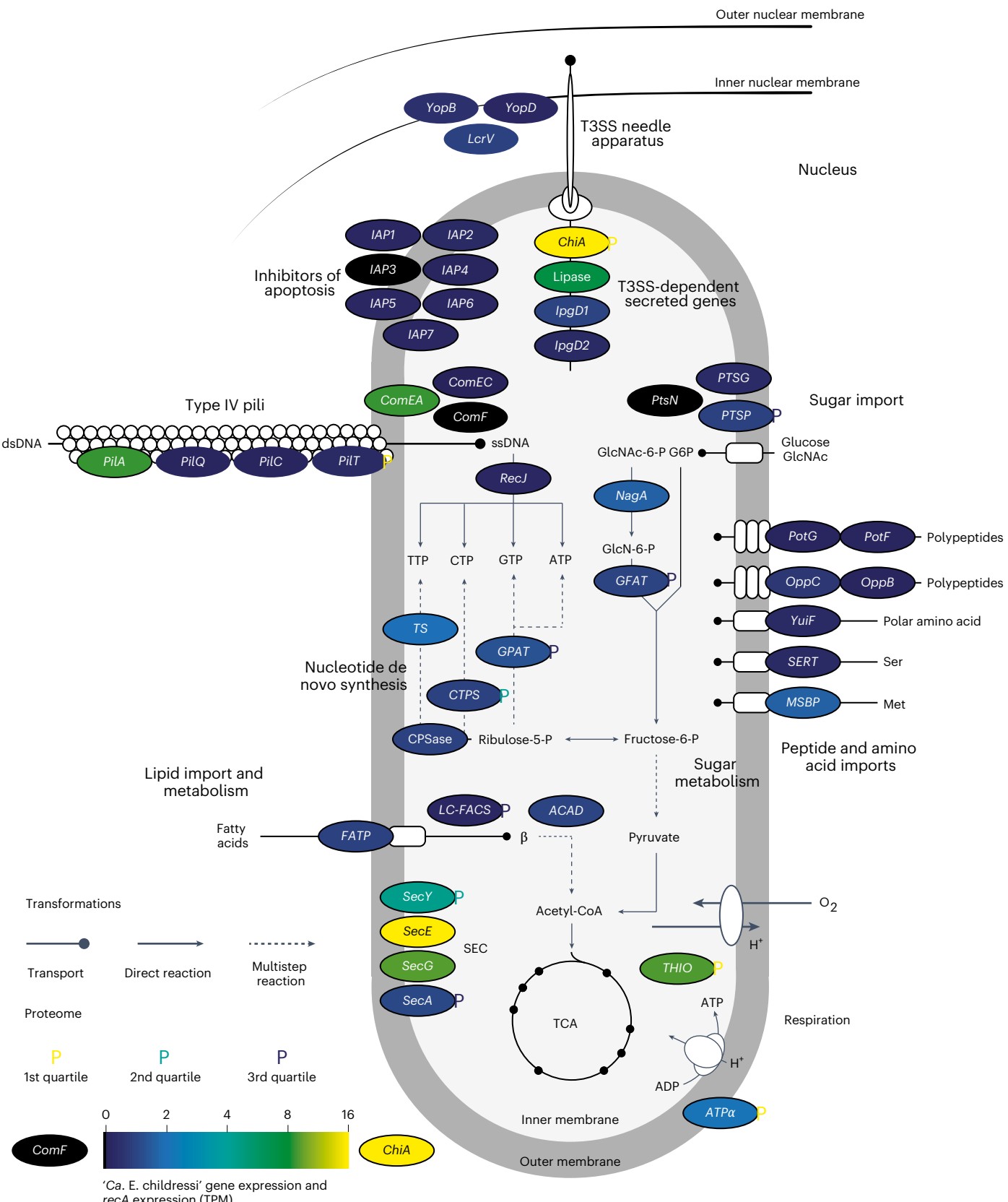

substrates that fuel the TCA-cycle-based core metabolism, and these were expressed in 'Ca. E. childressi'. The two parasites lacked synthesis pathways for amino acids (10 and 11 amino acids, respectively) (Fig. 1b and Supplementary Table 5), but encoded importers for polypeptides such as putrescine and importers for amino acids such as the generic importer *yuiF*, which were expressed in 'Ca. E. childressi' (Fig. 2 and Supplementary Tables 3 and 4).

Our dual RNA-seq analyses of laser-microdissected early, mid and late infection stages, as well as uninfected nuclei, revealed that 'Ca. E. childressi' expressed nutrient importers for sugars (carbohydrate

**Fig. 2 | 'Ca. E. childressi' is a chemoorganoheterotroph that gains its nutrition from lipids, sugars and amino acids from its host.** Physiological reconstruction from bulk tissue transcriptomes and proteomes. 'Ca. E. childressi' transcriptomic expression is shown in circled genes, as TPMs normalized to *recA* TPMs, for levels from not detected (black) through low (blue) to high (yellow) (colour legend on bottom left). The expression levels of proteins are shown as coloured 'P' symbols next to their corresponding gene, with yellow showing high abundance (first quartile), turquoise medium abundance (second quartile) and blue low abundance (third quartile). *ACAD*, acyl-CoA dehydrogenase; *ATPα*, ATP synthase alpha chain; *chiA*, chitinase; *comEA*, late competence protein ComEA DNA receptor; *comEC*, DNA internalization-related competence protein ComEC; *comF*, competence protein F homologue; *CPS*, carbamoyl-phosphate synthase large chain; *CTPS*, cytidine triphosphate synthase; *FATP*, long-chain fatty acid transport protein; *GFAT*, glucosamine-fructose-6-phosphate aminotransferase; *GPAT*, amidophosphoribosyltransferase; *ipgD1–2*, *Shigella*-like inositol phosphate phosphatases; *LC-FACS*, long-chain-fatty-acid-CoA ligase; *lcrV*, T3SS translocon protein LcrV; *LIP*, lipase; *MSBP*, methionine ABC transporter substrate-binding protein; *nagA*, *N*-acetylglucosamine-6-phosphate deacetylase; *oppB*, oligopeptide transport system permease protein OppB; *oppC*, oligopeptide transport system permease protein OppC; *pilA*, type IV pilin PilA; *pilC*, type IV fimbrial assembly protein PilC; *pilQ*, type IV pilus biogenesis protein PilQ; *pilT*, twitching motility protein PilT; *potF*, putrescine ABC transporter putrescine-binding protein PotF; *potG*, putrescine transport ATP-binding protein PotG; *PTSG*, glucose-specific component of PTS system; *ptsN*, nitrogen-regulatory protein of PTS system PtsN; *PTSP*, phosphoenolpyruvate-protein phosphotransferase of the PTS system; *recJ*, single-stranded-DNA-specific exonuclease RecJ; *secA*, protein export cytoplasm protein SecA ATPase RNA helicase; *secE*, preprotein translocase subunit SecE; *secG*, preprotein translocase subunit SecG; *secY*, preprotein translocase secY subunit; *SERT*, serine transporter; *THIO*, thioredoxin; *TS*, thymidylate synthase; *yuiF*, histidine permease YuiF; *yopB*, T3SS translocon protein YopB; *yopD*, T3SS translocon protein YopD; β, β-oxidation of fatty acids. For space reasons, only central aspects of 'Ca. E. childressi' metabolism and physiology are shown; the complete list of expressed genes in transcriptomic and proteomic analyses is available in Supplementary Tables 9 and 10. Signal peptide analyses of DNAses, RNAses and proteases are shown in Supplementary Table 11.

phosphotransferase system (*PTS*) genes), lipids (fatty acid transporter (*FATP*) genes) and amino acids (*yuiF*) throughout its infection cycle, with the highest upregulation during the early and mid infection stages (Figs. 2 and 3, and Supplementary Tables 3 and 7). Concomitantly, the host expressed genes for the import of sugars, amino acids and the synthesis of lipid droplets in the early and mid infection stages (Fig. 3 and Supplementary Table 8). In the late infection stage, the parasite decreased expression of genes involved in nutrient import, while the mussel decreased the expression of genes for importing sugar and synthesizing lipid droplets (Fig. 3 and Supplementary Tables 7 and 8). This could be because the host cell is no longer able to maintain its metabolism owing to nutrient depletion and/or 'Ca. Endonucleobacter' no longer grows considerably just before its release when the host cell bursts.

The two 'Ca. Endonucleobacter' species encoded a chitinase, a trait common to many Endozoicomonadaceae[21,22]. The chitinase was highly expressed in 'Ca. E. childressi', in both the bulk transcriptomes and proteomes, as well as in the laser-microdissected transcriptomes of all three infection stages (Figs. 2 and 3, and Supplementary Tables 3, 4 and 7). The chitinases of both 'Ca. Endonucleobacter' species encoded a signal peptide for type III secretion system (T3SS)-dependent secretion and were phylogenetically related to the *chiA*-2 chitinase of *Vibrio cholerae* (Extended Data Fig. 2). In *V. cholerae*, *chiA*-2 enables it to use mucin as a source of nutrition by deglycosylating mucin and releasing sugars such as *N*-acetylglucosamine (GlcNAc) and its oligomers[23]. If *chiA*-2 functions similarly in 'Ca. E. childressi', extracellular mucins of the mussel produced by the secretory cells of the gill would provide a rich source of nutrition. Mucin-derived sugars could be taken up by the mussel through its *SWEET* importer and degraded in the cytoplasm to GlcNAc by the chitobiase *CTBS*, as both genes were upregulated by the host in early and mid infection stages. The resulting cytoplasmic GlcNAc could then diffuse into the nucleus and be taken up by 'Ca. Endonucleobacter' via its phosphotransferase system *PTS*. What remains unclear

is how the parasite's chitinase is exported to the extracellular mucin. While speculative, it is possible that it is secreted by the T3SS through the nuclear envelope into the endoplasmic reticulum, and then exported via exocytosis through the host epithelial membrane to the extracellular mucin covering the gill cells.

'Ca. Endonucleobacter' is unlikely to use DNA, RNA and histones as its main source of nutrition based on the following evidence. When bacteria use DNA for nutrition, such as *V. cholerae* or *Escherichia coli*, they secrete DNAses to the extracellular medium or the periplasm, where the DNA is digested extracellularly, and oligonucleotides and monomers are then imported by the bacteria[24,25]. Both 'Ca. Endonucleobacter' species lacked nucleotide importers of the ADP/ATP translocase (TLC) family known from intranuclear bacteria of protists such as Rickettsiales, *Caedibacter caryophilus* and *Holospora* spp.[26–28]. Both 'Ca. Endonucleobacter' species lacked known genes for external secretion of DNAses or RNAses, and the few proteases that had secretion signal peptides were not expressed, were expressed at low levels or were restricted to the periplasm in 'Ca. E. childressi' (Supplementary Table 11). While 'Ca. E. childressi' expressed DNAses such as exodeoxyribonucleases I, III, V and VII and *recJ*, none of them were predicted to have secretion signal peptides, indicating that these DNAses are used for housekeeping tasks such as DNA replication and repair, and recycling. Also absent from both 'Ca. Endonucleobacter' species were nucleotide importers used by *V. cholerae* and *E. coli* to import DNA-derived nucleotides[24,25]. Instead, the two intranuclear parasites had all the genes for synthesizing their own nucleotides, and these were expressed in 'Ca. E. childressi' (Fig. 2 and Supplementary Tables 3, 4 and 12).

Both 'Ca. Endonucleobacter' species encoded competence factors for DNA import such as *comEA*, *comEC* and *comF*, but only *comEA* was highly expressed in 'Ca. E. childressi'; *comEC* was expressed at low levels and *comF* not at all (Fig. 2 and Supplementary Tables 3 and 4). Type IV pili (T4P) can also play a role in DNA uptake[29], and most T4P genes were expressed in 'Ca. E. childressi' (Fig. 2 and Supplementary Tables 3, 4

**Fig. 3 | *G. childressi* gill cells remained transcriptionally and metabolically active throughout the infection cycle.** In all three infection stages, IAPs were upregulated by the parasite 'Ca. E. childressi', while the host upregulated caspases, proteins involved in initiating apoptosis that are inhibited by IAPs. Transcriptomic profiling of metabolic and apoptotic interactions between 'Ca. E. childressi' (light grey) and *G. childressi* (dark grey) in early (top panel), mid (middle panel) and late (bottom panel) infection stages. 'Ca. E. childressi' gene expression is plotted as average (*n* = 3) TPMs normalized to *recA* TPMs. *G. childressi* gene expression is plotted as fold changes to the previous infection stage. Gene expression of *G. childressi* cells in the early stage of infection were compared with that of uninfected *G. childressi* cells. 'Ca. E. childressi' genes: *ACAD*; *chiA*; *GFAT*; *IAP1–7*; *ipgD1–2*; *FATP*; *LC-FACS*; *LIP*, probable lipase; *lcrV*; *MSBP*; *nagA*; *pilA*; *pilC*; *pilQ*; *pilT*; *PTSG*; *secA*; *secE*; *secG*; *secY*; *SERT*; *yopB*; *yopD*; *yuiF*. *G. childressi* genes: *AGPAT*, 1-acyl-sn-glycerol-3-phosphate acyltransferase alpha; *ATPα*; *ATPβ*, ATP synthase beta chain; *CASP2-i1–15*, caspase-2 isoforms 1–15; *CTBS*, chitobiase; *DGAT*, diacylglycerol *O*-acyltransferase; *FAS*, fatty acid synthase; *FBPA*, fructose–bisphosphate aldolase; *GFAT*; *GPAT*, glycerol-3-phosphate *O*-acyltransferase; *GPDH*, glycerol-3-phosphate dehydrogenase; *nagA*; *PAP*, phosphatidate phosphatase; *PFK-1*, phosphofructokinase-1; *PLIN2*, perilipin-2; *PKM*, pyruvate kinase PKM; *rBAT*, neutral and basic amino acid transport protein rBAT; *SCNA*, sodium-coupled neutral amino acid transporter; *SWEET*, SWEET sugar transporter 1; *SYNE1*, Nesprin-1. Not all genes involved in glycolysis, the TCA cycle and β are shown for space reasons, but are listed in Supplementary Tables 14 and 15.

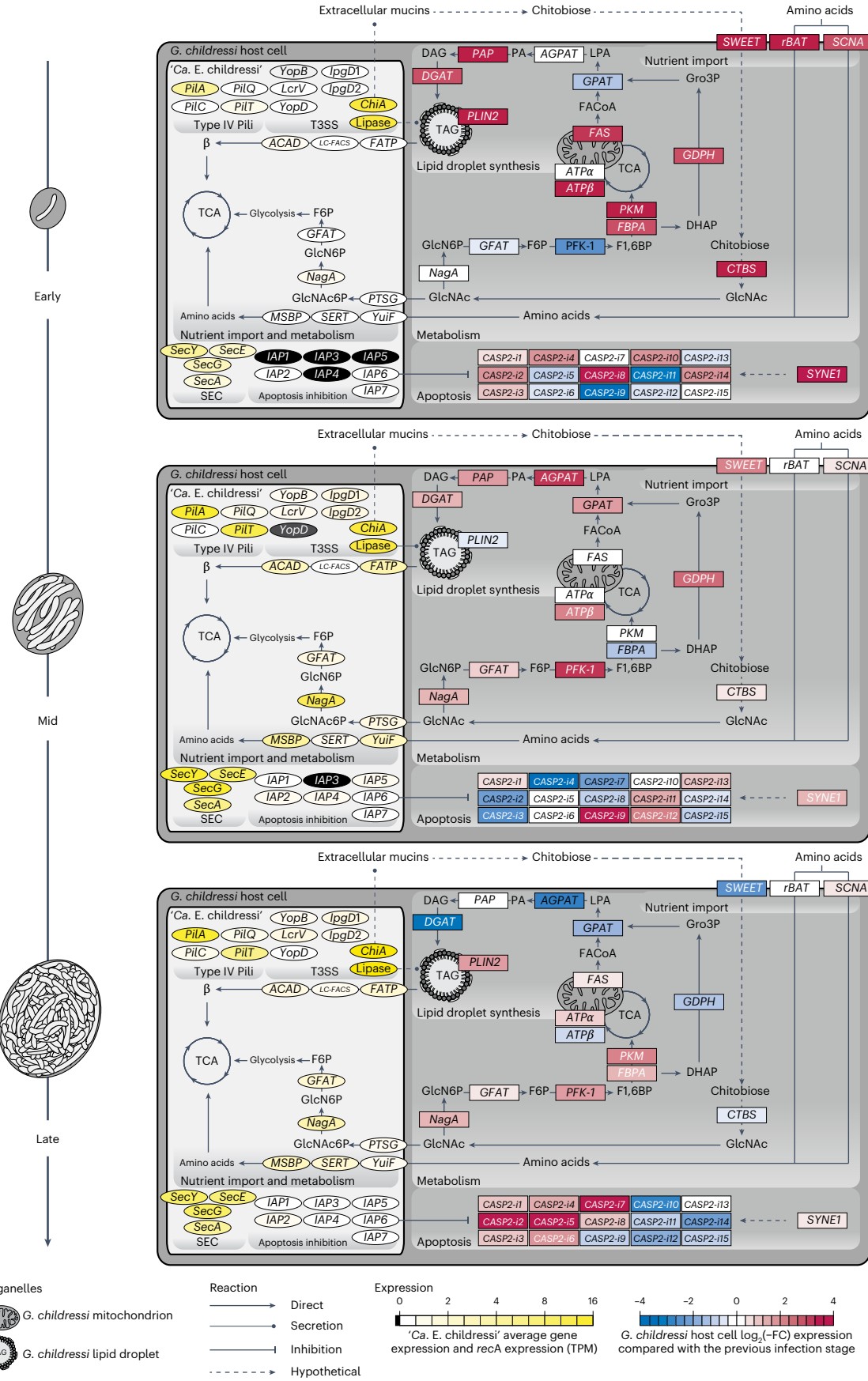

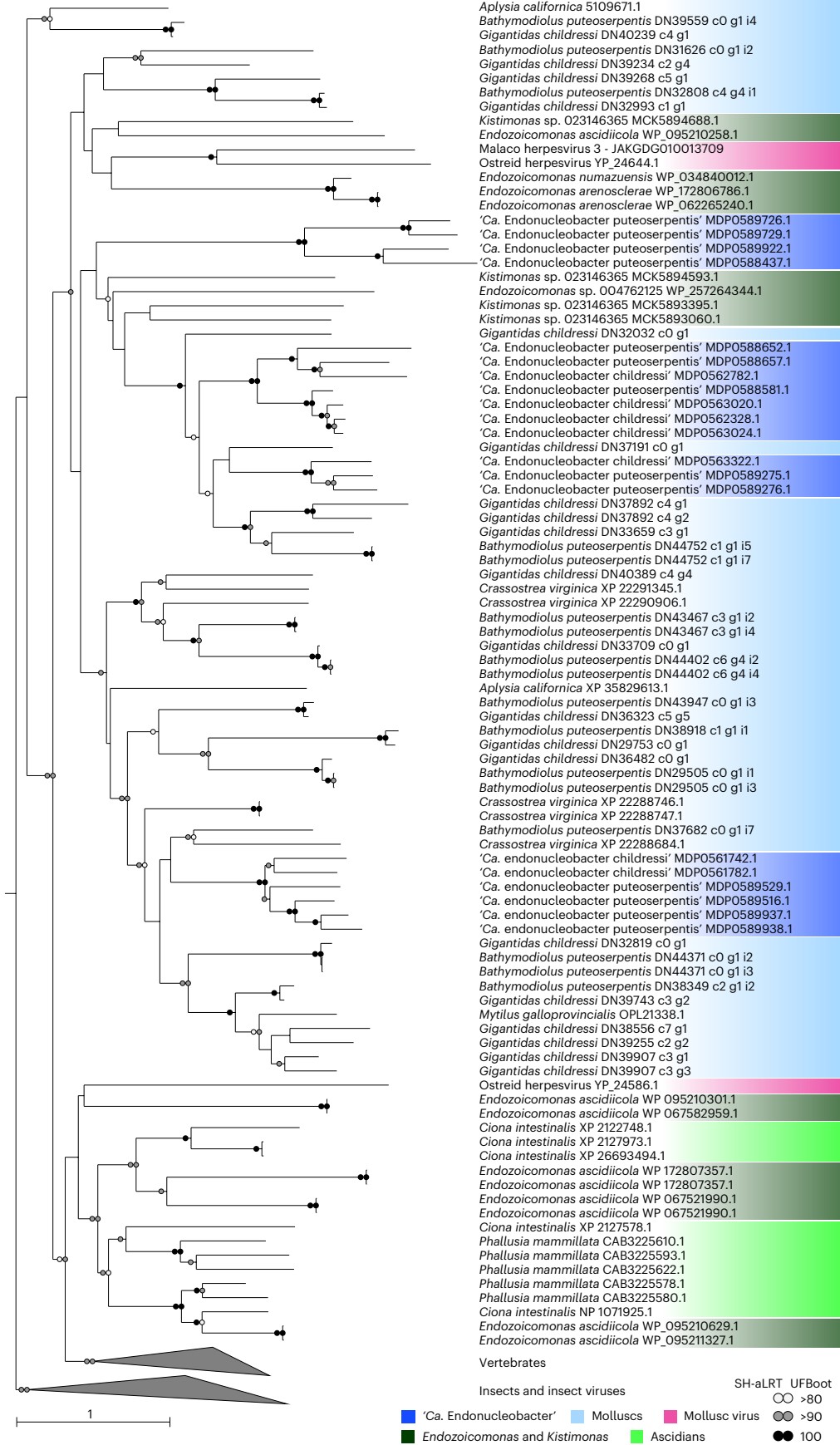

**Fig. 4 | IAPs encoded by 'Ca. Endonucleobacter' are interspersed with those of their mussel host and viruses from the family Malacoherpesviridae.** Protein-based phylogeny of 128 IAPs from 'Ca. Endonucleobacter', *Endozoicomonas*, *Kistimonas*, metazoans and viruses. Sequences were aligned with MAFFT, the tree calculated with IQ-TREE and branch support (1,000 replicates) calculated with both SH-aLRT and UFBoot tree. Bootstrap values ≥80 are shown. The scale bar indicates substitutions per site. Colours indicate taxonomic groups. Insects and their virus IAPs were included to root the tree.

and 13). However, T4P is also known to play roles in facilitating adherence to host cells, surface movement (twitching motility), phage adsorption and biofilm formation[29].

In summary, while we cannot exclude that T4P and competence genes could be involved in DNA uptake in 'Ca. Endonucleobacter', the lack of expression of genes involved in secreting nucleases and importing nucleotides, together with the high expression of genes involved in the digestion of sugars, lipids and amino acids, indicates that DNA is not the main source of nutrition for 'Ca. E. childressi' (Fig. 2 and Supplementary Tables 3 and 4).

On the host side, our analyses of different infection stages provided further evidence that 'Ca. E. childressi' does not appear to consume considerable amounts of host DNA, RNA or histones (Supplementary Table 8). We found no evidence for downregulation of host transcription, as expected if nuclear DNA and RNA were consumed[12]. The infected host cell remained transcriptionally and metabolically active throughout the infection cycle, as host genes involved in glycolysis and oxidative phosphorylation were expressed, even in late-stage nuclei (Fig. 3 and Supplementary Table 8). Moreover, light and electron microscopy analyses revealed that the mussel host cells remained morphologically asymptomatic, apart from the swollen nucleus, with intact membranes and organelles (Extended Data Figs. 3 and 4).

### 'Ca. Endonucleobacter' and its host engage in an apoptotic arms race

One highly unusual feature of 'Ca. Endonucleobacter' is that its genome encodes inhibitors of apoptosis (IAPs), with 'Ca. E. childressi' and 'Ca. E. puteoserpentis' encoding 7 and 13 IAPs, respectively (Supplementary Table 2 and Supplementary Note 2). IAPs are an evolutionarily conserved group of proteins that are common to animals and have been horizontally acquired by some invertebrate viruses[30], but have not been previously described in bacteria[31]. In animals, IAPs inhibit a process of programmed cell death called apoptosis, mainly by binding caspases, proteases that play a central role in inducing apoptosis[32]. IAP proteins contain one to three baculoviral IAP repeat (BIR) motifs that allow them to sequester caspases[33], and are therefore sometimes referred to as BIR-containing proteins (BIRPs). Only BIRPs that have a RING domain, which can ubiquinate caspases to target them for proteolysis via the proteasome, are considered bona fide apoptosis inhibitors[34]. In our analysis, we therefore refer to proteins as IAPs only if they had both a BIR and a RING domain (Extended Data Fig. 5).

To understand the role IAPs play in the biology of 'Ca. Endonucleobacter', we studied the genes expressed by both the parasite and its host during the infection cycle (Figs. 2 and 3, and Supplementary Tables 3, 4, 7 and 8). All seven IAPs encoded by 'Ca. E. childressi' had signal peptides for the Sec secretion pathway, and the genes secA, secY, secE and secG were expressed in all infection stages (Figs. 2 and 3, and Supplementary Tables 3, 4 and 7). 'Ca. E. childressi' first expressed three IAPs in early infection stages, six in mid stages, and finally all seven IAPs in late stages of infection (Fig. 3 and Supplementary Table 7). Concomitantly, the host expressed as many as 16 different caspase-2 isoforms throughout the infection cycle (Fig. 3 and Supplementary Table 8).

A wide range of stimuli can trigger apoptosis, such as metabolic stress, DNA damage and ER stress. One of the caspases that initiates the apoptotic cascade is caspase-2 (ref. 35). IAPs are well known for their ability to bind to and block caspases[33]. Although not well studied in marine invertebrates, in the oyster Crassostrea gigas, IAP-2 strongly binds to and blocks caspase-2, suggesting that IAPs play an important role in the inhibition of caspase-2-mediated apoptosis in bivalves[36]. While secretion of bacterial IAPs into the nucleus could appear counterintuitive, caspase-2 has been shown to localize to, and induce apoptosis from, the nucleus, for example, as a reaction to DNA damage[37].

The concomitant upregulation of G. childressi caspase-2 isoforms and 'Ca. E. childressi' IAPs during the infection cycle suggests that the host initiates apoptosis in response to the infection, swelling of

its nucleus and hijacking of its metabolism (Supplementary Notes 3 and 4), which 'Ca. E. childressi' counters by upregulating IAPs. Thus, both the host and the intranuclear parasite engage in a physiological arms race for control of apoptosis, with seven different IAPs of 'Ca. E. childressi' preventing an arsenal of G. childressi caspase isoforms from inducing apoptosis long enough for the parasite to acquire the energy and nutrients it needs to replicate to such high numbers before the death of its host cell.

### 'Ca. Endonucleobacter' acquired IAPs from its host

Although IAPs have not been previously reported from bacterial genomes, our analyses revealed that in addition to 'Ca. Endonucleobacter', four Endozoicomonas and one Kistimonas species from other marine invertebrates also encoded bona fide IAPs (Supplementary Table 2). Comparative phylogenetic analyses of 'Ca. Endonucleobacter' and other Endozoicomonadaceae IAPs with publicly available animal and viral IAPs revealed that bacterial IAPs were not monophyletic, but rather fell into nine clades that were interspersed with IAPs of marine invertebrates (Fig. 4). IAPs in all three 'Ca. Endonucleobacter' clades were most closely related to those of their bathymodioline hosts, as well as other molluscs (Fig. 4). Similarly, most bacterial IAPs from another group, namely, Endozoicomonas ascidiicola isolated from ascidians, were most closely related to ascidian IAPs (Fig. 4 and Supplementary Note 5). Two viral IAPs, from the ostreid herpesvirus OsHV-1 and from MalacoHV3 (Malacoherpesviridae family)[38], were also interspersed between those of Endozoicomonadaceae and molluscs (Fig. 4). OsHV-1, first found in oysters[39], infects a wide range of molluscs[40]. This virus also appears to infect bathymodioline mussels. We recovered 17% of the OsHV-1 genome in the same G. childressi specimen from which we assembled the 'Ca. E. childressi' genome, providing evidence that 'Ca. E. childressi' and OsHV-1 coexist in the same host individual.

The interspersed phylogeny of Endozoicomonadaceae IAPs with those of marine invertebrate and viral IAPs suggests horizontal gene transfer (HGT) of these genes between animals, bacteria and viruses. Given that caspase-mediated apoptosis is specific to animals, and IAPs are known only from animals and some viruses that acquired IAPs horizontally from their invertebrate hosts, IAPs in bacteria are likely not ancestral, but were rather acquired through HGT from animals or viruses. HGT from animals to bacteria has only rarely been observed, although it is common between bacteria, and from bacteria to eukaryotes, particularly in protists[41,42]. As to how 'Ca. Endonucleobacter' acquired its host's IAPs, three mutually non-exclusive explanations are plausible: (1) 'Ca. Endonucleobacter' acquired the IAPs via its competence genes or T4P for taking up DNA (Fig. 2). (2) HGT of IAPs could have also been facilitated by the numerous mobile genetic elements in 'Ca. Endonucleobacter', with insertion sequences constituting 10–11% of the parasite's genome (Supplementary Table 2). (3) Viruses could act as vectors for HGT between kingdoms. Support for this third explanation stems from our transmission electron microscopy (TEM) analyses, which revealed viral-like structures inside 'Ca. E. childressi' whose structure resembles double-stranded DNA viruses (Extended Data Fig. 6)[39]. OsHV-1 is a double-stranded DNA virus that reportedly infects the nuclei of the oyster Crassostrea gigas[43], but without evidence that the viral-like structures we observed in 'Ca. E. childressi' are from OsHV-1, this scenario remains purely speculative. While direct evidence for viruses that can infect both prokaryotes and eukaryotes is lacking, in some cases of HGT from bacteria to eukaryotes, phages have been proposed to be the vectors for HGT between kingdoms, such as insects that encode bacterial toxins closely related to orthologues from bacteriophages of insect symbionts[44].

### Discussion

The presence of IAPs in an intranuclear bacterial parasite poses the chicken or the egg question. Did the intranuclear lifestyle of 'Ca. Endonucleobacter' allow the acquisition of IAPs from its host, or did the

acquisition of IAPs allow '*Ca.* Endonucleobacter' to make a living in the nucleus? If colonization of the nucleus by the parasite occurred first before acquiring IAPs, then division rates of ancestral '*Ca.* Endonucleobacter' would have had to have been low enough to not induce apoptosis. Alternatively, if '*Ca.* Endonucleobacter' acquired IAPs before its intranuclear lifestyle, this would have required at least two steps: (1) intimate contact with the host's DNA or mRNA or that of a virus of the host and (2) a certain frequency of this contact. Both requirements would suggest an intracellular lifestyle with the ability to invade the nucleus at least occasionally. *Rickettsia* that infect insects have such a lifestyle, as they are generally intracellular but occasionally invade their host's nuclei[12]. The presence of IAPs in *Endozoicomonas*, some of which cluster with the IAPs from their ascidian host, raises the question whether these *Endozoicomonas* could also occasionally invade their host's nuclei. To date, most Endozoicomonadaceae associations have, however, not been analysed with imaging methods so that this question cannot be currently resolved. Beyond Endozoicomonadaceae, our database queries recovered a few bacterial IAP sequences in samples collected from soils, the deep sea and the phyllosphere, but again, imaging analyses would be needed to reveal whether these sequences originated from bacteria associated with eukaryotes (Supplementary Table 18).

Not all intracellular bacteria use IAPs to avoid apoptosis[45]. For example, the intranuclear bacteria that colonize protists do not have IAPs, based on our database queries. However, protists lack bona fide caspases, and their apoptosis is caspase independent[46,47]. Moreover, if a bacterium that lives in a unicellular host is passed on to both daughter cells, it could lead a sheltered, intranuclear lifestyle as long as the bacterium does not have major negative effects on its host's fitness. Indeed, some of the intranuclear bacteria that colonize protists can be benign[48,49]. By contrast, the intranuclear bacteria of deep-sea mussels colonize terminally differentiated cells and must therefore reproduce before their host cell dies, explaining the strong selective advantage in acquiring IAPs.

Our study adds to the small but growing body of evidence for HGT from eukaryotes to bacteria[41,50–52]. HGT from eukaryotes to prokaryotes is assumed to be disfavoured for several reasons including the presence of eukaryotic introns as barriers to genetic transfer of genes to prokaryotes, and the lack of eukaryotic metabolic versatility compared with bacteria[41]. Eukaryotes have numerous genes and pathways for interacting with both beneficial and parasitic bacteria. Acquisition of these genes by bacteria could improve their ability to enter and reproduce in their eukaryotic hosts, as argued here for the acquisition of IAPs by '*Ca.* Endonucleobacter'. One of the most striking examples for the selective advantage of having eukaryotic-like proteins is *Legionella pneumophila*, which acquired a large number of these proteins as effectors for interfering with host pathways, mainly from the protists they infect[53]. Similarly, work on sponges identified eukaryotic-like proteins in their symbionts that mediate phagocytosis[54,55]. These examples, together with our study, indicate that HGT from eukaryotes to bacteria may be more common than currently recognized, particularly in bacteria that are closely associated with eukaryotic hosts[56]. As large-scale sequencing efforts aimed at a holistic view of the genomic underpinnings of eukaryotic organisms and their associated microbiome are now becoming more common, we are in an ideal position to revisit our understanding of eukaryote-to-prokaryote HGT events.

## Methods

### Sample collection

*G. childressi* mussels were collected using the remotely operated vehicle (ROV) Hercules during the RV Meteor Nautilus NA-58 cruise to the Gulf of Mexico in May 2015 at the Mississippi Canyon site (MC853; 28° 7′ N, 89° 8′ W) and the Green Canyon site (GC234; 27° 45′ N, 91° 13′ W) at water depths of 1,070 m and 540 m, respectively. *B. puteoserpentis* mussels were collected using the ROV MARUM-QUEST during the

Meteor M126 cruise to the Mid-Atlantic Ridge in April 2016 from the Logatchev vent field (Irina-II smoker; 14° 45′ N, 44° 59′ W) at a water depth of 3,036 m. Onboard, the mussels' gills were dissected, preserved and stored as described below. Samples for DNA sequencing, bulk RNA sequencing and poly(A)-RNA sequencing were preserved in RNAlater (Thermo Fisher Scientific) and stored at −80 °C. Samples for microscopy and laser-capture microdissection transcriptomic analyses were fixed in 4% paraformaldehyde in 1× PBS for 8 h at 4 °C and stored in 0.5× PBS–50% ethanol at −20 °C. Samples for proteomics were snap-frozen with liquid nitrogen and stored at −80 °C. Samples for TEM were fixed in 2.5% glutaraldehyde in PHEM buffer (piperazine-*N*,*N*′-bis,4-(2-hydroxyethyl)-1-piperazineethanesulfonic acid, ethylene glycol-bis (β-aminoethyl ether) and MgCl$_2$ for 12 h at 4 °C (ref. [57])) and then stored in PHEM buffer. Metadata for the collected specimens are in Supplementary Table 1.

### Microscopy

**Fluorescence microscopy.** Whole-filament overviews (Extended Data Fig. 1a,d) were visualized with the epifluorescence microscope Olympus BX53 (Olympus) with a UCPlanFL 20X/0.70 air transmission lens and an Orca Flash 4.0 camera (Hamamatsu) using the Olympus cellSens Dimension software v. 1.18 (Olympus). Detailed images (Fig. 1a and Extended Data Figs. 1b,c,e,f and 4a–c) were recorded with a Zeiss LSM 780 equipped with an Airyscan detector and two different objectives, a plan-APROCHROMAT 63×/1.4 oil immersion objective and a plan-APROCHROMAT 100×/1.46 DIC M27 Elyra oil immersion objective. Images were obtained and post-processed using ZEN software (black edition, 64 bits, version 14.0.1.201, Carl Zeiss Microscopy). Images were adjusted for brightness and levels using the software Adobe Photoshop (version 12.0, Adobe Systems).

***G. childressi* 18S rRNA levels based on fluorescence signal intensities.** To investigate whether infection of mussel gill cells by '*Ca.* Endonucleobacter' led to a reduction in rRNA amounts, we measured the 18S rRNA fluorescence intensity of uninfected gill cells, and compared these to early, mid and late stages of infection (Extended Data Fig. 4d). Gill filaments from *G. childressi* specimen H1423/001-N5-002 were hybridized as described in 'Whole-mount FISH' with the probe BNIX64 specific for '*Ca.* Endonucleobacter' and the eukaryotic EUK-1195 probe[22] (Extended Data Fig. 4a–c), and relative fluorescence intensity measured in ten areas of identical surface per cell using Fiji 1.52v[23].

**Whole-mount FISH.** Gill filaments of *G. childressi* (H1423/002-N9) and *B. puteoserpentis* (499ROV/1-4) were dissected and hybridized for 3 h at 46 °C with 500 nM of oligonucleotide probes targeting 16S rRNA (Supplementary Table 6) in hybridization buffer containing 35% formamide, 80 mM NaCl, 400 mM Tris–HCl, 0.4% blocking reagent for nucleic acids (Roche), 0.08% SDS (v/v) and 0.08 dextran sulfate (w/v). Following hybridization, the gill filaments were washed in pre-warmed 48 °C washing buffer (0.07 M NaCl, 0.02 M Tris–HCl (pH 7.8), 5 mM EDTA (pH 8) and 0.01% SDS (v/v)) for 15 min. After washing, the gill filaments were counterstained with DAPI for 10 min at room temperature, transferred to poly-L-lysine-coated glass slides (Sigma-Aldrich), mounted overnight at room temperature using the ProLong Gold anti-fade mounting media (Thermo Fisher Scientific) and stored at −20 °C until visualization.

**TEM.** For TEM analyses, gill tissues were post fixed with 1% (v/v) osmium tetroxide (OsO$_4$) for 2 h at 4 °C, washed three times with PHEM and dehydrated in an ethanol series (30%, 50%, 70%, 80%, 90% and 100% (v/v)) at −10 °C for 10 min each. Tissues were transferred to 50:50 ethanol and acetone, followed by 100% acetone, and infiltrated with low-viscosity resin (Agar Scientific) using centrifugation embedding[58]. Samples were centrifuged for 30 s in resin:acetone mixtures of 25%, 50%, 75% and twice in 100%, transferred into fresh resin in embedding

moulds and polymerized at 60–65 °C for 48 h. Ultrathin (70 nm) sections were cut on a microtome (Ultracut UC7 Leica Microsystem), mounted on formvar-coated slot grids (Agar Scientific) and contrasted with 0.5% aqueous uranyl acetate (Science Services) for 20 min and with 2% Reynold's lead citrate for 6 min. Sections were imaged at 20–30 kV with a Quanta FEG 250 scanning electron microscope (FEI Company) equipped with a scanning transmission electron microscopy detector using the xT microscope control software v6.2.6.3123.

## Proteomics

**Proteomic sample preparation and liquid chromatography with tandem mass spectrometry analysis.** We dissected the ciliated edges of mussel gills, which are enriched in '*Ca*. Endonucleobacter', from snap-frozen gills of 13 *G. childressi* specimens (Supplementary Table 1). For tryptic protein digestion, the filter-aided sample preparation (FASP) protocol, adapted from ref. 59, was used. Depending on the amount of tissue, 100 μl or 150 μl of SDT-lysis buffer (4% (w/v) SDS, 100 mM Tris–HCl (pH 7.6), 0.1 M DTT) was added and samples were heated at 95 °C for 10 min. To minimize sample loss, we omitted the 5-min centrifugation step at 21,000 *g* as described in the original FASP protocol and, instead, only briefly spun down the homogenate for a few seconds. The remainder of the FASP protocol and determination of peptide concentrations were done as described in ref. 60. For each liquid chromatography with tandem mass spectrometry (LC–MS/MS) run, 1,500 ng of peptide was loaded onto a 5-mm, 300-μm-internal diameter C18 Acclaim PepMap100 pre-column (Thermo Fisher Scientific) using an UltiMate 3000 RSLCnano Liquid Chromatograph (Thermo Fisher Scientific) and desalted on the pre-column. The pre-column was switched in line with a 75 μm × 75 cm analytical EASY-Spray column packed with PepMap RSLC C18, 2 μm material (Thermo Fisher Scientific), which was heated to 55 °C. The analytical column was connected via an Easy-Spray source to a Q Exactive HF-X Hybrid Quadrupole-Orbitrap mass spectrometer (Thermo Fisher Scientific). Peptides were separated on the analytical column using a 460 min gradient as described in ref. 61. Mass spectra were acquired in the Orbitrap as described in ref. 62 with some modifications. Briefly, eluting peptides were ionized via electrospray ionization and analysed in Q Exactive HF-X. Full scans were acquired in the Orbitrap at 60,000 resolution. The 15 most abundant precursor ions were selected for fragmentation, isolated with the quadrupole using a 1.2 m/z window, fragmented in the higher-energy collisional dissociation cell with 25 normalized collision energy and measured at 7,500 resolution. Singly charged ions were excluded and dynamic exclusion was set to 30 s. On average, 258,842 MS/MS spectra were acquired per sample.

**Proteomics data processing.** We built a protein sequence database from the '*Ca*. E. childressi' genome (*G. childressi* specimen H1423/002-N9) and common laboratory contaminants using the cRAP protein sequence database v2012.01.01 (http://www.thegpm.org/crap/) (Supplementary Table 10). We searched the MS/MS spectra against this database using Sequest HT in Proteome Discoverer version 2.2.0.388 (Thermo Fisher Scientific) as described previously in ref. 2. Proteins were filtered to achieve a false discovery rate <5%. For protein quantification, normalized spectral abundance factors[63] were calculated per species. A subset of the detected proteins was used to supplement the metabolic model of '*Ca*. E. childressi' (Fig. 2 and Supplementary Table 4).

## DNA and RNA extraction

**DNA extraction and screening for '*Ca*. Endonucleobacter'.** We PCR screened 15 *G. childressi* and 5 *B. puteoserpentis* RNAlater-preserved gill samples for '*Ca*. Endonucleobacter' (Supplementary Table 1). DNA was extracted using the DNeasy Blood and Tissue Kit (Qiagen) following the manufacturer's protocol. The '*Ca*. Endonucleobacter' 16S rRNA gene was PCR amplified using Taq DNA Polymerase (5 PRIME), with the

following conditions: initial denaturation for 3 min at 95 °C, 30 cycles at 95 °C for 30 s, 55 °C for 30 s and 72 °C for 2 min, followed by a final elongation step at 72 °C for 10 min. The '*Ca*. Endonucleobacter' 16S rRNA gene was amplified using the forward primer BNIX64 (AGCGGTAACAG-GTCTAGC)[10] and the reverse primer GM4 (TACCTTGTTACGACTT)[64].

**Metagenomic library preparation and sequencing.** We sequenced the DNA of one *G. childressi* individual (H1423/002-N9) and one *B. puteoserpentis* individual (499ROV/1-4) using short-read (Illumina HiSeq 3000) and long-read (PacBio) sequencing at the Max Planck Genome Center Cologne, Germany (https://mpgc.mpipz.mpg.de/home/). For short-read sequencing, 50 ng of genomic DNA was fragmented via sonication (Covaris S2, Covaris), followed by library preparation with NEBNext Ultra DNA v2 Library Prep Kit for Illumina (New England Biolabs). Library preparation included seven cycles of PCR amplification. Quality and quantity were assessed at all steps via capillary electrophoresis (TapeStation, Agilent Technologies) and fluorometry (Qubit, Thermo Fisher Scientific). The library was immobilized and processed onto a flow cell with cBot (Illumina) and subsequently sequenced on a HiSeq 3000 system (Illumina) with 2 × 150 bp paired-end reads, to generate a total of 333 million paired-end reads. Long-read sequencing was done according to the manual 'Procedure and Checklist−20 kb Template Preparation Using BluePippin Size Selection' of Pacific Biosciences without initial DNA fragmentation and without a final size selection. Instead, libraries were purified twice with PB AMPure beads. Sequencing was performed on a Sequel device with Sequel Binding Kit 3.0 and Sequel Sequencing Kit 3.0 for 20 h (Pacific Biosciences). A total of two and three sequencing PacBio cells were generated for *G. childressi* and *B. puteoserpentis*, respectively.

***G. childressi* de novo transcriptome.** To study host cell expression throughout the infection cycle, we assembled a *G. childressi* transcriptome de novo. We dissected the ciliated edges of 20 RNAlater-preserved gill filaments from *G. childressi* H1423/002/N6. RNA was extracted and prepared as described in the next section with the following modifications: 1 μg of total RNA was used for library preparation, poly(A) enrichment was done with the NEBNext poly(A) mRNA Magnetic Isolation Module (New England Biolabs) and library preparation with the NEBNext Ultra II Directional RNA Library Prep Kit for Illumina (New England Biolabs), and 11 cycles of PCR amplification, generating a total of 99 million paired-end reads.

**Bulk transcriptomics.** We dissected the ciliated edges of nine RNAlater-preserved gill filaments from *G. childressi* specimen H1423/002-N9 and extracted total RNA using the RNeasy Mini Kit (Qiagen, Germany) following the manufacturer's protocol. RNA quantity was measured with a QUANTUS Fluorometer (Promega, Germany). Library preparation and sequencing was performed as described in 'Metagenomic library preparation and sequencing' for the short-read library preparation, with the following modifications: 20 ng of total RNA was used for library preparation, and libraries prepared using the NEBNext Ultra II Directional RNA Library Prep Kit for Illumina (New England Biolabs), generating a total of 33 million paired-end reads.

## Laser-capture microdissection

**Dissection of infected *G. childressi* nuclei.** We used the formalin-fixed gills of *G. childressi* H1423/002/N6 for laser-capture microdissection. Gill filaments were embedded in polyester wax, sectioned at 10 μm using a microtome and mounted on thermoexitable polyester membranes (number 115005191, Leica). Sections were hybridized with the '*Ca*. Endonucleobacter' 16S rRNA probe BNIX64 as described above with the following modifications: the hybridization buffer did not contain formamide, only the '*Ca*. Endonucleobacter' 16S rRNA was used, sections were not DAPI stained and no mounting medium was used after air-drying. A Leica LMD6500 (Leica) was used to dissect

the hybridized samples. Per infection stage, 100 nuclei were microdissected and pooled in a single tube prefilled with 30 µl of extraction buffer (AllPrep DNA/RNA FFPE kit, Qiagen). In addition, 100 uninfected nuclei were dissected as just described to establish a baseline for host expression. For each of the three infection stages (early, mid and late) as well as uninfected nuclei, triplicates were prepared, resulting in a total of 1,200 microdissected nuclei from 12 samples.

**Laser-capture microdissection transcriptomics.** *RNA extraction and sequencing.* We extracted RNA from the microdissected nuclei using the AllPrep DNA/RNA FFPE kit (Qiagen) following the manufacturer's protocol with the following modifications: samples were incubated in proteinase K overnight at 37 °C, the elution buffer was pre-warmed at 37 °C and added to the column membrane, and the incubation time in the elution buffer doubled. After a first elution step, the eluent was loaded on the membrane again, incubated for 2 min and eluted again. RNA quantity was assessed with a Quantus Fluorometer (Promega). Library preparation and sequencing was done as described in 'Metagenomic library preparation and sequencing' for the short-read library preparation, with the following modifications: total RNA was amplified following the protocol of capture and amplification by tailing and twitching described in ref. 65. Libraries were prepared with the RNA-seq Kit v2 (Diagenaode) and included 16 cycles of PCR amplification, with 150 bp single-end reads sequenced. To obtain similar amounts of '*Ca*. E. childressi' mRNA reads in each library, we adjusted the number of reads sequenced per library according to '*Ca*. E. childressi' mRNA abundance, detailed in Supplementary Table 16.

*Expression analysis.* Expression analysis of laser-capture microdissection (LCM) RNA reads was done as described in 'Expression analysis of '*Ca*. E. childressi'', with these additions for the analysis of the host cell: we removed non-mRNA contaminants and bacterial contaminants by mapping the reads against rRNA and tRNA SILVA database v132 (ref. 66) and against the genomes of '*Ca*. E. childressi' and the methane-oxidizing symbiont of *G. childressi* using BBMap v38.90 (https://sourceforge.net/projects/BBMap) (identity: 0.85). After removal of contaminants, LCM reads were mapped against the *G. childressi* de novo transcriptome using BBMap v38.90 (identity: 0.85). Mapped reads were counted with FeatureCounts v1.6.3 (ref. 67) and analysed using Aldex2 v3.11 (ref. 68) in RStudio v1.3.1093 (ref. 69) considering the different infection stages (uninfected, early, mid and late) as conditions. Fold changes in expression between consecutive stages were calculated at 128 Monte Carlo instances and using the median abundance of all features as a denominator for the geometric mean calculation (Supplementary Table 15). Fold changes of *G. childressi* gene expression at an early stage of infection were calculated in base to the uninfected *G. childressi* cells. '*Ca*. E. childressi' gene expression per infection stage was quantified by calculating transcripts per million (TPMs) normalized to *recA* (Supplementary Table 14). A subset of the expression data was used to reconstruct the infection interactions shown in Fig. 3 (Supplementary Tables 7 and 8). The variation of the expression data for the parasite and host was calculated using vegan v2.6-4 (ref. 70) in RStudio v1.3.1093 and visualized as a non-metric multidimensional scaling plot (Supplementary Fig. 2).

### Bioinformatic analyses
**Genome assembly.** Short reads were screened for '*Ca*. Endonucleobacter' using phyloFlash v3.3 (ref. 71) and assembled using Spades v3.7 (ref. 72) after decontamination, quality filtering (trimq = 2) and adaptor trimming using BBDuk v38.90 (sourceforge.net/projects/BBMap v38.90/). We binned '*Ca*. E. childressi' and '*Ca*. E. puteoserpentis' draft genomes from their respective '*G. childressi*' and '*B. puteoserpentis*' metagenomes using Gbtools v2.6.0 (ref. 73). '*Ca*. E. childressi' and '*Ca*. E. puteoserpentis' short-read genomes were assembled using the draft genomes as references by using BBMap v38.90 (identity: 0.98)

and Spades v3.7 (maximum k-mer size of 127)[72]. We eliminated contigs shorter than 1 kb from the short-read genomes, screened for contamination using Bandage v0.8.1 (ref. 74) and checked for quality metrics using CheckM v1.0.18 (ref. 75). A '*Ca*. E. puteoserpentis' high-quality draft metagenome-assembled genome was assembled by mapping long reads against the '*Ca*. E. puteoserpentis' short-read genome using ngmlr v.0.2.7 (ref. 76) and assembled using CANU v2.0 (ref. 77). The assembled long reads were supplemented with the short-read genome using Unicycler v0.4.8 (ref. 78). The '*Ca*. E. childressi' high-quality draft metagenome-assembled genome was assembled from PacBio HiFi long reads using CANU. The '*Ca*. E. childressi' genome was extracted from the graphic representation of the CANU assembly using its 16S rRNA gene as a bait in Bandage. We eliminated contigs shorter than 1 kb from the genomes and checked for quality metrics using CheckM v1.0.18. The genomes were classified as high-quality draft metagenome-assembled genomes according to the quality standards established in ref. 79. We annotated the genomes using RAST v2.0 (ref. 80) and cross-checked RAST annotations manually using v.2.10.1 NCBI BLAST.

**Expression analysis of '*Ca*. E. childressi'.** We quality trimmed the RNA reads and removed adaptors with BBDuk v38.90. Reads were mapped against the rRNA and tRNA SILVA database v132 (ref. 66) using BBMap v38.90 (identity: 0.85) to remove non-mRNA contaminants. We quantified the expression of '*Ca*. E. childressi' using Kallisto v.0.44.0 (ref. 81) with default settings (Supplementary Table 9). Transcription levels were normalized to the single-copy housekeeping gene *RecA* (Supplementary Table 3) and mapped onto metabolic pathways using Pathway tools v13.0 (ref. 82) for reconstruction of '*Ca*. E. childressi' metabolism in Fig. 2.

**Assembly, curation and annotation.** Poly(A) RNA reads were quality trimmed and adaptors were removed using BBDuk v38.90. To remove bacterial contaminants, we mapped reads against the genomes of '*Ca*. E. childressi' and the methane-oxidizing symbiont. Non-mRNA reads from other potential bacterial contaminants were removed by mapping against the rRNA and tRNA SILVA database using BBMap v38.90 (identity: 0.85). After decontamination, reads were normalized with BBNorm v38.90 (https://sourceforge.net/projects/BBMap) and assembled with Spades v3.7. We checked the preliminary assembly for completeness and quality metrics using the Trinity Stats package from Trinity v.2.10.0 (ref. 83) and BUSCO v.4.1.2 (metazoan database)[84], and taxonomic affiliations assigned to the reads of the preliminary assembly using BLAST. Reads were uploaded into MEGAN v.6.16.4 (ref. 85) and non-eukaryotic reads removed from the preliminary assembly. The resulting assembly was annotated using the Trinotate package from Trinity v.2.10.0.

**IAP database search.** We used the hmm profile we generated for identifying IAPs and queried the UniProt database for bacterial IAPs, using the hmmsearch webserver at https://www.ebi.ac.uk/Tools/hmmer/search/hmmsearch (ref. 86). Positive hits were checked to verify the presence of both a BIR and a RING domain, and are listed in Supplementary Table 18.

**Metabolic reconstruction.** To reconstruct the metabolism of '*Ca*. E. childressi', we loaded its RAST-annotated genome into Pathway tools v13.0 (ref. 82). We interpreted the metabolism of '*Ca*. E. childressi' from the bulk transcriptome and proteome analyses of *G. childressi* H1423/002-N9 (Fig. 2 and Supplementary Tables 3 and 4). SignalP v6.0 (ref. 87) was used for signal peptide analysis (Supplementary Table 11).

**Phylogenomics and comparative genomics.** We analysed the phylogeny of 172 single-copy genes shared between the two '*Ca*. Endonucleobacter' genomes and those of 42 closely related Endozoicomonaceae (Supplementary Table 2). We used the GToTree v1.8.4 program[88] to download representative Endozoicomonadaceae genomes from GTDB

Release 09-RS220 (ref. [89]), identified the 172 single-copy gene set with HMMER3 v3.4 (ref. [90]), aligned the single-copy genes with Muscle 5.1.linux64 (ref. [91]) and trimmed the alignment with TrimAl v1.4.rev15 (ref. [92]). IQ-TREE 2.3.0 was used for tree calculations. The percentage of insertion sequences relative to the total genome content of each genome was analysed using ISEScan[93].

**IAP identification and verification.** We identified BIRPs in 'Ca. E. childressi', 'Ca. E. puteoserpentis' and related Endozoicomonadaceae genomes by conducting a protein homology analysis. We aligned a total of 48 publicly available BIRP amino acid sequences from tunicates, vertebrates, molluscs, arthropods, entomopoxviruses and Malacoherpesviridae using MAFFT v7.407 (refs. [26,27]). From the BIRP alignment, we generated a hidden Markov model using the hmmbuild function from hmmer v3.1b2 (ref. [28]) and screened genomes using the hmmsearch function of hmmer at default thresholds ($E$ value $1 \times 10^{-3}$). Candidate BIRPs were analysed for functional protein domains using the NCBI online service for protein domain prediction (https://www.ncbi.nlm.nih.gov/Structure/cdd/wrpsb.cgi). We classified BIRPs with both BIR repeats and RING domains as bona fide IAPs (Extended Data Fig. 5). To verify that the identified IAPs were not contaminants from the mussel host, we visualized the assembly graph of the 'Ca. E. childressi' genome using Bandage v0.8.1 and, using the inbuilt BLAST function, ensured that the IAPs originated from the bacterial contigs (Supplementary Fig. 1).

**IAP phylogeny.** A total of 7 amino acid IAP sequences from 'Ca. E. childressi', 13 from 'Ca. E. puteoserpentis', 9 from *Endozoicomonas ascidiicola*, 2 from *Endozoicomonas arenosclerae*, 1 from *Endozoicomonas numazuensis*, 1 from *Endozoicomonas* sp. ONNA2 and 1 from Endozoicomonadaceae bacterium SW310 (that is, *Kistimonas* sp.) were aligned using MAFFT v7.407, together with 20 *G. childressi* host IAP sequences annotated in this study, 17 *B. puteoserpentis* host IAPs annotated in ref. [94] and 59 publicly available IAP sequences from tunicates, vertebrates, molluscs, arthropods, entomopoxviruses, ostreid herpesviruses and malacoherpesviruses. The phylogenetic tree was reconstructed using the maximum-likelihood-based software IQ-TREE v2.3.0 using ModelFinder (VT + I + R5 substitution model, with 1,000 replicates for the ultrafast bootstrap and 1,000 replicates for the SH-like approximate likelihood ratio test)[95]. Before inclusion in the analysis, all putative IAPs from public databases were subjected to the same checks mentioned in Supplementary Note 2.

**Chitinase phylogeny and protein domain analyses.** We inferred the phylogeny of the 'Ca. Endonucleobacter' chitinases by comparing them to 38 chitinase amino acid sequences from the G18 glycosidase family. All sequences were aligned using MAFFT v7.471. The phylogenetic tree was reconstructed using the maximum-likelihood-based software IQ-TREE v1.6.12 using the TIM3 substitution model (1,000 bootstraps). Protein domain analysis of the 'Ca. E. childressi' chitinase was done using the NCBI online service for protein domain prediction against the CDD v.3.21-62456 PSSMs database (https://www.ncbi.nlm.nih.gov/Structure/cdd/wrpsb.cgi).

### Statistics and reproducibility

No statistical methods were used to pre-determine sample sizes, but our sample sizes are similar to those reported in previous publications[96,97]. The experiments were not randomized. Data collection and analysis were not performed blind to the conditions of the experiments. One of the LCM samples of the 'uninfected' group was excluded from further analysis as preparation of the sequencing library failed.

### Reporting summary

Further information on research design is available in the Nature Portfolio Reporting Summary linked to this article.

### Data availability

The metagenomic and metatranscriptomic raw reads and assembled symbiont genomes are available in the National Center for Biotechnology Information (NCBI) under BioProject accession number PRJNA979916. The annotated genomes of both 'Ca. Endonucleobacter' species, as used in this study, the host transcriptomes and their annotations, the HMM profiles used to identify IAPs, and the microscope data used to generate the figures are available via Zenodo at https://doi.org/10.5281/zenodo.11086255 (ref. [98]). The mass spectrometry metaproteomics data and protein sequence database were deposited in the ProteomeXchange Consortium via the PRIDE[99] partner repository with the dataset identifier PXD020317.

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

## Acknowledgements

We are grateful to the captains, crew, ROV teams and chief scientists of the research cruises RV Nautilus NA-58 (2015) and RV Meteor M126 (2016) for their support. We thank S. Wetzel, A. Ellrott and D. Tienken from the Max Planck Institute for Marine Microbiology for technical assistance and B. Hüttel and L. Czaja-Hasse from the Max Planck Genome-Centre Cologne for the sequencing and the protocol optimization. This study was funded by the Max Planck Society, a European Research Council Advanced Grant to N.D. (BathyBiome, 340535), a Gordon and Betty Moore Foundation Marine Microbial Initiative Investigator Award to N.D. (grant GBMF3811), the Gottfried Wilhelm Leibniz Prize of the German Research Foundation (DFG) to N.D., the USDA National Institute of Food and Agriculture Hatch project 1014212 (M.K.) and the US National Science Foundation (grants OIA 1934844 and IOS 2003107 to M.K.). LC–MS/MS measurements

were made in the Molecular Education, Technology, and Research Innovation Center (METRIC) at North Carolina State University. H.G.-V. was partially funded by the DFG (Heisenberggrant GR 5028/1-1).

## Author contributions

M.Á.G.P., N.D. and N.L. conceived the project and designed the experiments. M.Á.G.P. and N.L. carried out microscopy experiments. M.Á.G.P., A.A., H.G.-V. and N.L. performed genome sequencing and analyses. M.Á.G.P. and A.A. identified and analysed IAPs. M.Á.G.P. and N.L. designed, optimized and performed the LCM pipeline. M.Á.G.P., M.T., H.G.-V. and N.L. performed transcriptome sequencing and analyses. M.V. and M.K. performed proteomic data generation and analyses. M.Á.G.P., H.G.-V. and N.L. performed phylogenetic analyses. M.Á.G.P., N.D. and N.L. wrote the paper with input from all co-authors.

## Funding

## Competing interests

The authors declare no competing interests.

## Additional information

**Extended data** is available for this paper at https://doi.org/10.1038/s41564-024-01808-5.

**Correspondence and requests for materials** should be addressed to Nicole Dubilier or Nikolaus Leisch.

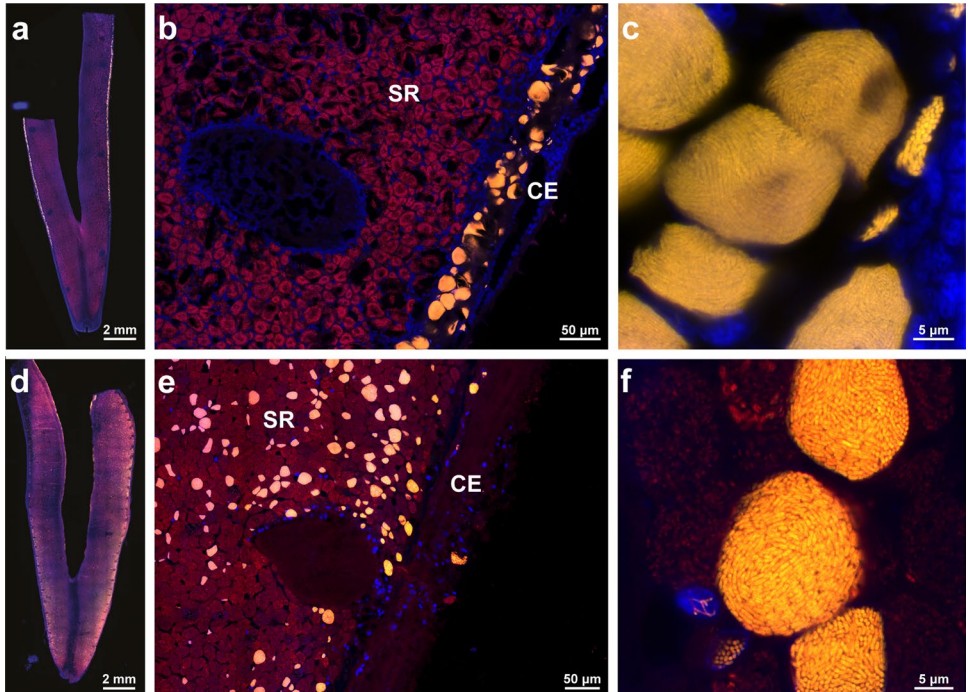

**Extended Data Fig. 1 | Distribution pattern of '*Ca*. Endonucleobacter' in the host tissue.** '*Ca*. E. childressi' only colonized nuclei of cells in the ciliated edges of *G. childressi* gills, while '*Ca*. E. puteoserpentis' infected the nuclei of cells throughout the gill tissues of *B*. *puteoserpentis*. *a-f*, Fluorescence *in situ* hybridization (FISH) micrographs of single gill filaments of *G. childressi* (**a-c**) and *B. puteoserpentis* specimens (**d-f**) with the FISH probe specific to '*Ca*. Endonucleobacter' shown in yellow, the eubacterial probe in red, and DAPI-stained DNA in blue. **a** and **d** show stitched overviews of whole gill filaments. **b** and **e** show the symbiotic region (SR) with the sulfur- and methane-oxidizing symbionts, and the ciliated edge (CE), and **c** and **f** show infected nuclei at higher resolution.

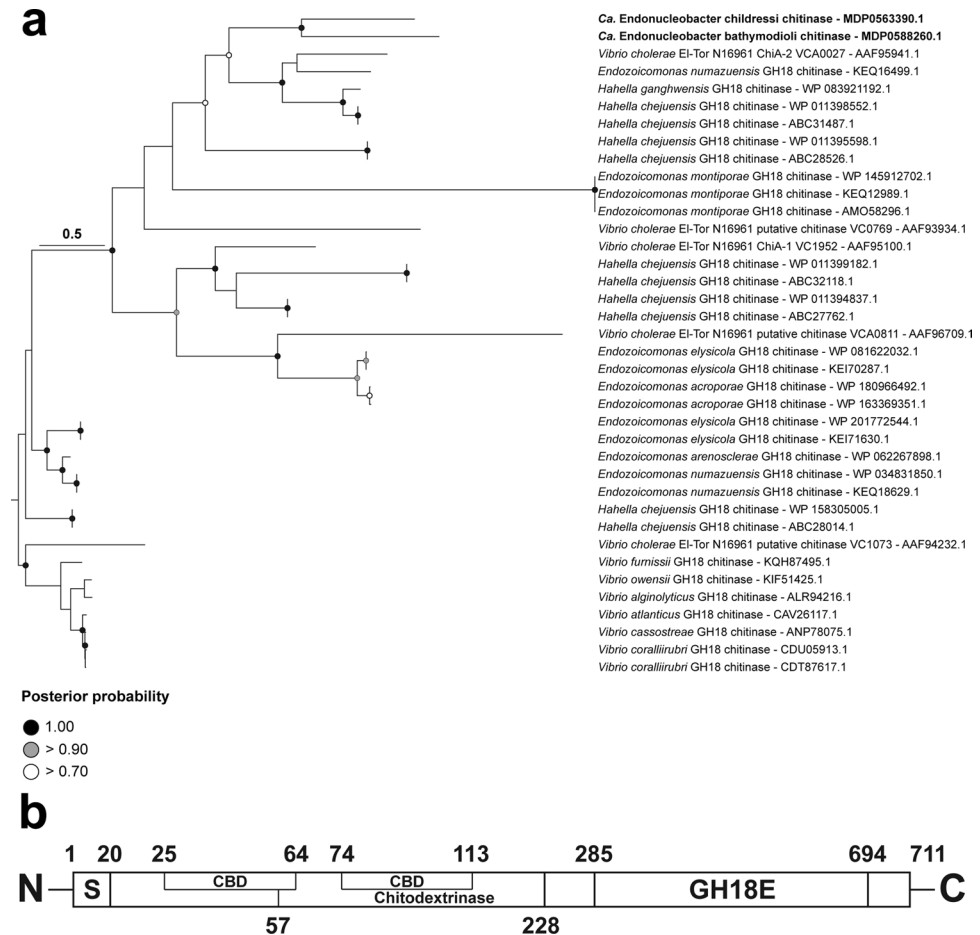

**Extended Data Fig. 2 | Chitinase phylogeny and domain analysis.**
'*Ca*. **Endonucleobacter**' chitinases are related to *Vibrio cholerae* chitinase
**ChiA-2. The** '*Ca*. **E. childressi' chitinase has a N-terminal peptide for**
**secretion via a T3SS. (a)**, Protein-based phylogeny of 38 MAFFT v7.471-aligned
chitinases from '*Ca*. Endonucleobacter', *Endozoicomonas*, *Hahella* and
*Vibrio* calculated using the maximum likelihood-based software IQTREET

v1.6.12 (TIM3 substitution model, bootstrap: 1000). We used seven chitinase
sequences from *Vibrio* spp. representatives to root the tree. **(b)**, NCBI (https://
www.ncbi.nlm.nih.gov/Structure/cdd/wrpsb.cgi) protein domain analysis of
'*Ca*. E. childressi' chitinase (**S**, T3SS secretion signal peptide; **CBD**, chitin-binding
domain; **Chitodextrinase**, chitodextrinase domain; **GH18E**, catalytic domain).
Numbers indicate the domain position in the amino acid sequence (not scaled).

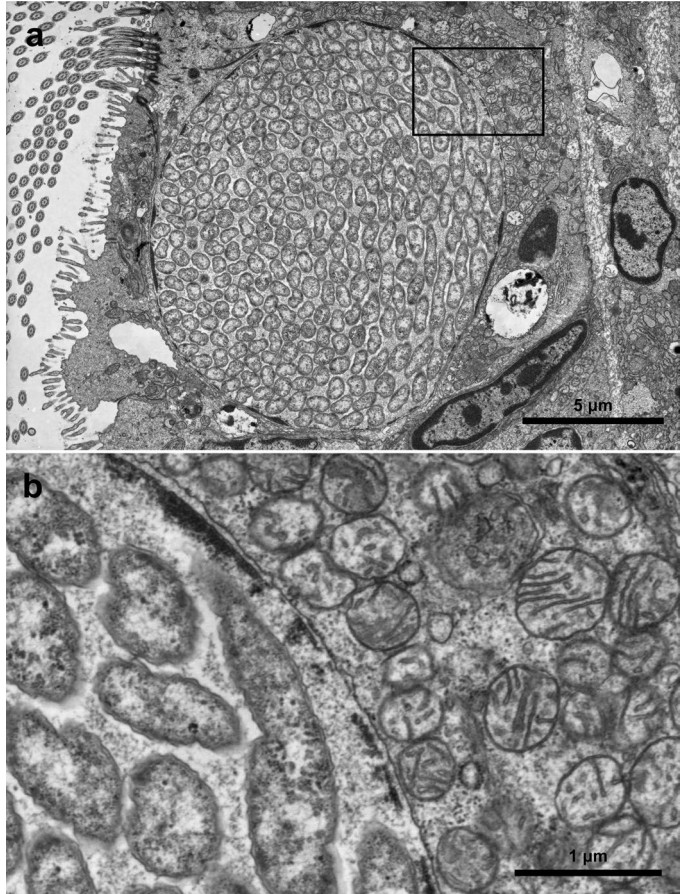

**Extended Data Fig. 3 | Ultrastructure of a cell infected by 'Ca. E. childressi'. Late infection stage of 'Ca. E. childressi'. Mitochondria, membranes and other cellular features of the host cell are morphologically intact. (a)**, TEM overview of a G. childressi cell infected by 'Ca. E. childressi'. **(b)**, Higher resolution micrograph of rectangle in **(a)** showing host chromatin compressed along the inner nuclear membrane and morphologically intact mitochondria in the cytosol.

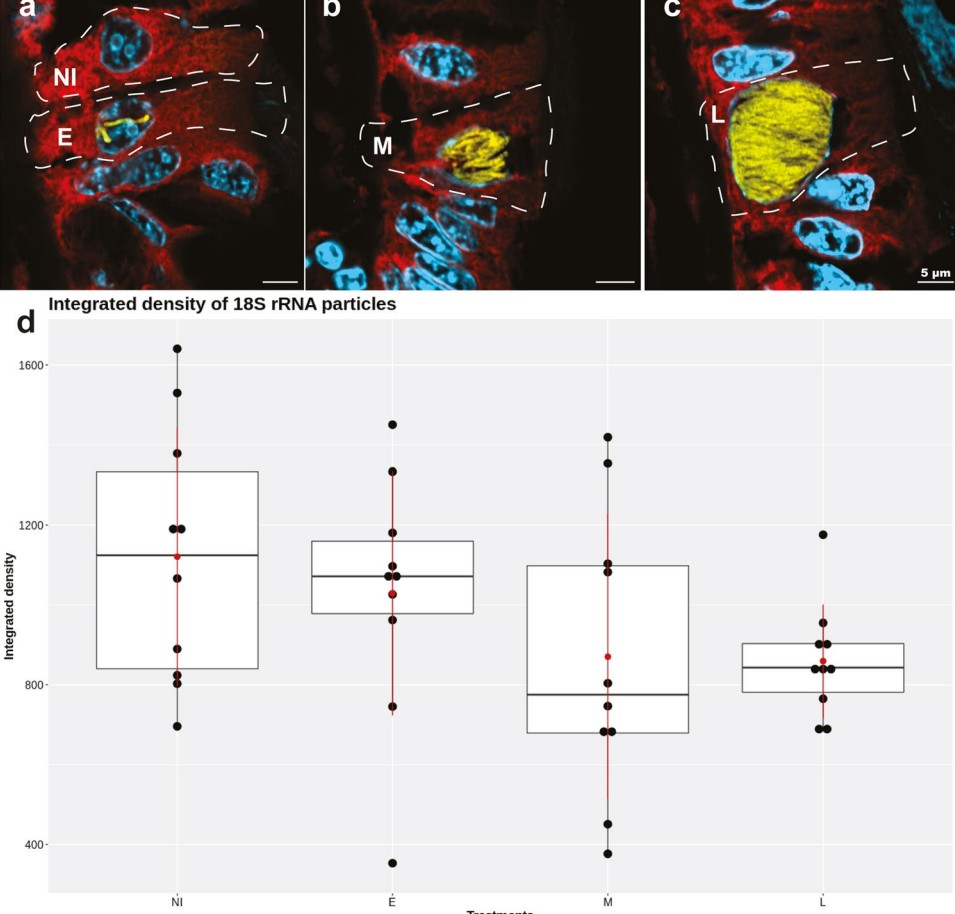

**Extended Data Fig. 4 | Fluorescence intensity analysis of 18S rRNA in** ***G. childressi*** **throughout the infection. The fluorescence intensity of host 18S rRNA in** ***G. childressi*** **cells did not change substantially during the infection cycle.** We measured the relative fluorescence intensity of host 18S rRNA in ten areas of equivalent surface per cell using Fiji [127] as an indicator of host transcription and ribosomal activity. **(a) - (c):** Fluorescence *in situ* hybridization (FISH) micrographs of gill cells at different '*Ca*. E. childressi' infection stages: **(a)** non-infected and early-stage of infection (**NI** = not infected, **E** = early).

**(b)** mid-stage of infection (**M** = mid). **(c)** late-stage of infection (**L** = late) with the FISH probe specific to '*Ca*. Endonucleobacter' shown in yellow, the eukaryotic probe in red, and DAPI-stained DNA in cyan (sequences of all FISH probes are listed in Supplementary Table 6). **(d)**, integrated fluorescence intensity of host 18S rRNA normalized to respective cell area at different '*Ca*. E. childressi' infection stages (n = 10). Black dots represent data points, black lines within boxes represent medians, boxes indicate 25–75 percentiles, red dots represent means and whiskers denote standard deviation.

## *Ca*. E. childressi IAPs

**IAP1 (QS721_05160)**

N— ... BIR ... BIR ... RING ... —C

**IAP2 (QS721_05385)**

N— BIR ... BIR ... BIR ... BIR ... BIR ... RING —C

**IAP3 (QS721_08390)**

N— ... BIR ... BIR ... RING ... —C

**IAP4 (QS721_10855)**

N— ... BIR ... BIR ... RING ... —C

**IAP5 (QS721_12130)**

N— ... BIR ... BIR ... RING ... —C

**IAP6 (QS721_12150)**

N— ... BIR ... BIR ... RING ... —C

**IAP7 (QS721_13745)**

N— ... BIR ... BIR ... RING ... —C

## *Ca*. E. puteoserpentis IAPs

**IAP1 (QS748_08830)**

N— ... BIR ... BIR ... RING ... —C

**IAP2 (QS748_08840)**

N— ... BIR ... BIR ... RING ... —C

**IAP3 (QS748_10140)**

N— BIR ... BIR ... BIR ... BIR ... BIR ... BIR ... RING ... —C

**IAP4 (QS748_10205)**

N— ... BIR ... BIR ... BIR ... RING ... —C

**IAP5 (QS748_11270)**

N— ... BIR ... RING ... —C

**IAP6 (QS748_11285)**

N— ... BIR ... RING ... —C

**IAP7 (QS748_12370)**

N— ... BIR ... RING ... —C

**IAP8 (QS748_12450)**

N— BIR ... BIR ... BIR ... BIR ... BIR ... RING ... —C

**IAP9 (QS748_12455)**

N— ... BIR ... BIR ... RING ... —C

**IAP10 (QS748_04310)**

N— ... BIR ... RING ... —C

**IAP11 (QS748_05070)**

N— ... BIR ... BIR ... RING ... —C

**IAP12 (QS748_05425)**

N— ... BIR ... BIR ... RING ... —C

**IAP13 (QS748_05450)**

N— ... BIR ... BIR ... RING ... —C

**Extended Data Fig. 5 | IAPs domain analysis. '*Ca*. E. childressi' and '*Ca*. E. puteoserpentis' encoded for seven and 13 bona fide IAPs respectively, with BIR and RING domains**. Domain analysis of all IAPs encoded by '*Ca*. E. childressi' on the left, and '*Ca*. E. puteoserpentis' on the right (**BIR**, BIR-repeats domain; **RING**, RING domain). Numbers indicate the domain position in the amino acid sequence (not scaled).

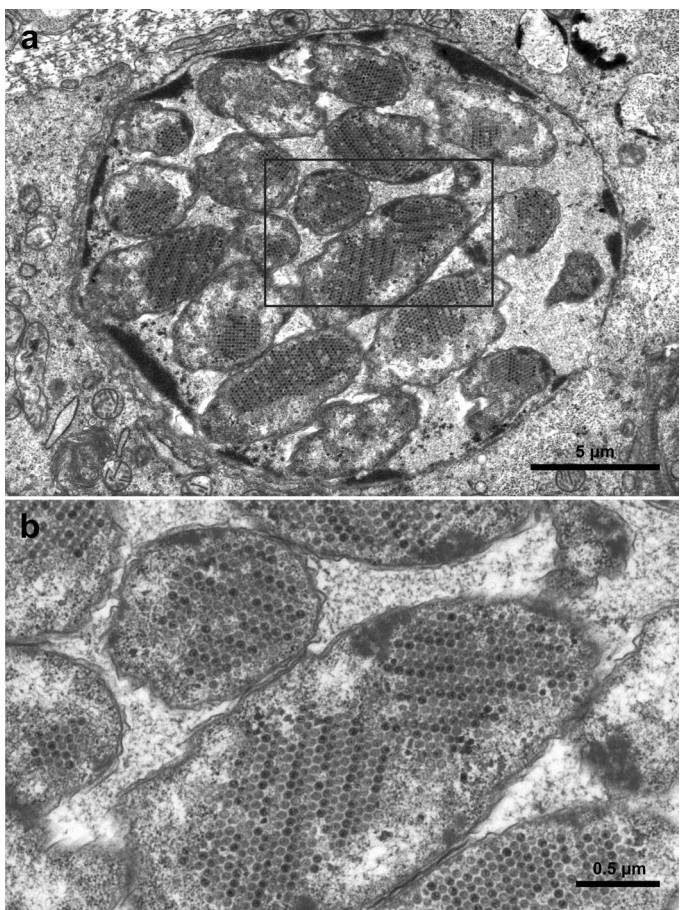

**Extended Data Fig. 6 | Viral infection of *Ca*. Endonucleobacter cells. '*Ca*. E. childressi' had structures typical for viruses in their cells. (a)**, TEM image of *G. childressi* gill cell showing a nucleus infected by '*Ca*. E. childressi' that is filled with structures resembling icosahedral viral capsids. **(b)**, Higher magnification of rectangle in (a) showing the putative viral capsids inside '*Ca*. E. childressi' cells. As we wrote in our manuscript, we did find OsHV-1 sequences (17% of the genome) in a library from a single Bathymodiolus individual. However, we obviously cannot show that these sequences originated from the virus-like particles we observed in our TEM images.

# Reporting Summary

## Statistics

For all statistical analyses, confirm that the following items are present in the figure legend, table legend, main text, or Methods section.

| n/a | Confirmed | |
|---|---|---|
| ☐ | ☒ | The exact sample size (*n*) for each experimental group/condition, given as a discrete number and unit of measurement |
| ☐ | ☒ | A statement on whether measurements were taken from distinct samples or whether the same sample was measured repeatedly |
| ☒ | ☐ | The statistical test(s) used AND whether they are one- or two-sided<br>*Only common tests should be described solely by name; describe more complex techniques in the Methods section.* |
| ☒ | ☐ | A description of all covariates tested |
| ☒ | ☐ | A description of any assumptions or corrections, such as tests of normality and adjustment for multiple comparisons |
| ☐ | ☒ | A full description of the statistical parameters including central tendency (e.g. means) or other basic estimates (e.g. regression coefficient) AND variation (e.g. standard deviation) or associated estimates of uncertainty (e.g. confidence intervals) |
| ☒ | ☐ | For null hypothesis testing, the test statistic (e.g. *F*, *t*, *r*) with confidence intervals, effect sizes, degrees of freedom and *P* value noted<br>*Give P values as exact values whenever suitable.* |
| ☒ | ☐ | For Bayesian analysis, information on the choice of priors and Markov chain Monte Carlo settings |
| ☒ | ☐ | For hierarchical and complex designs, identification of the appropriate level for tests and full reporting of outcomes |
| ☒ | ☐ | Estimates of effect sizes (e.g. Cohen's *d*, Pearson's *r*), indicating how they were calculated |

*Our web collection on statistics for biologists contains articles on many of the points above.*

## Software and code

Policy information about availability of computer code

| | |
|---|---|
| Data collection | No specialized software was used for data collection. |
| Data analysis | RAST v2.0: https://rast.nmpdr.org/<br>JGI annotation server: https://genome.jgi.doe.gov/portal/<br>NCBI BLAST v2.10.1<br>Pathway Tools v13.0<br>SignalP v6.0<br>MAFFT v7.407<br>MAFFT v7.471<br>IQTREE v1.6.12<br>xT microscope control software v6.2.6.3123<br>cRAP protein sequence database v2012.01.01: http://www.thegpm.org/crap/<br>Proteome Discoverer v2.2.0.388<br>BBMap v.38.90, including BBDuk, BBNorm https://sourceforge.net/projects/bbmap/<br>FeatureCounts v1.6.3<br>Aldex2 v3.11<br>PhyloFlash v3.3<br>Spades v3.7<br>Gbtools v2.6.0<br>Bandage v0.8.1<br>CheckM v1.0.18 |

ngmlr v0.2.7
CANU v2.0
Unicycler v0.4.8
SILVA database v123: https://www.arb-silva.de/
Kallisto v.0.44.0
Trinity v.2.10.0
BUSCO v.4.1.2
MEGAN v.6.16.4
NCBI protein domain analysis, CDD v.3.21-62456 PSSMs: https://www.ncbi.nlm.nih.gov/Structure/cdd/wrpsb.cgi
Fiji 1.52v
Clustal Omega v1.2.2
ISEScan  v1.7.2.3
Olympus cellSens Dimension software v1.18
ZEN software v14.0.1.201
Adobe Photoshop / Adobe Illustrator v12
GToTree v1.8.4
HHMMER3 v3.1b2
HHMMER3 v3.4
Muscle 5.1.linux64
TrimAl v1.4.rev15
Prodigal v2.6.3
Genome Taxonomy Database (GTDB), Release 09-RS220
FastTree 2 v2.1.11
GNU Parallel v20240122
IQTREE 2.3.0
RStudio v1.3.1093
vegan v2.6-4

For manuscripts utilizing custom algorithms or software that are central to the research but not yet described in published literature, software must be made available to editors and reviewers. We strongly encourage code deposition in a community repository (e.g. GitHub). See the Nature Portfolio guidelines for submitting code & software for further information.

# Data

Policy information about availability of data

 All manuscripts must include a data availability statement. This statement should provide the following information, where applicable:

- Accession codes, unique identifiers, or web links for publicly available datasets
- A description of any restrictions on data availability
- For clinical datasets or third party data, please ensure that the statement adheres to our policy

The metagenomic and metatranscriptomic raw reads and assembled symbiont genomes are available in The National Center for Biotechnology Information (NCBI) under BioProject Accession Number PRJNA979916. The annotated genomes of both "Ca. Endonucleobacter" species, as used in this study, the host transcriptomes and their annotations, and the HMM profiles used to identify IAPs and the microscope data used to generate the figures are available in the ZENODO repository under DOI: 10.5281/zenodo.11086255.
The mass spectrometry metaproteomics data and protein sequence database were deposited in the ProteomeXchange Consortium via the PRIDE partner repository with the dataset identifier PXD020317. The genomes of "Ca. Endonucleobacter childressi" and "Ca. Endonucleobacter puteoserpentis" generated in this study were submitted to NCBI under the accession numbers GCA030674875.1 and GCA030674915.1, respectively.
For the construction of the phylogenetic tree in Figure 1, genomes were downloaded from NCBI ftp://ftp.ncbi.nlm.nih.gov/genomes/all/ with accession numbers listed in Figure 1. For the construction of the phylogenetic tree in Figure 4, protein sequences were downloaded from NCBI https://ncbi.nlm.nih.gov/ with accession numbers listed in Figure 4.

# Research involving human participants, their data, or biological material

Policy information about studies with human participants or human data. See also policy information about sex, gender (identity/presentation), and sexual orientation and race, ethnicity and racism.

| Reporting on sex and gender | N/A |
|---|---|
| Reporting on race, ethnicity, or other socially relevant groupings | N/A |
| Population characteristics | N/A |
| Recruitment | N/A |
| Ethics oversight | N/A |

Note that full information on the approval of the study protocol must also be provided in the manuscript.

# Field-specific reporting

Please select the one below that is the best fit for your research. If you are not sure, read the appropriate sections before making your selection.

☐ Life sciences  ☐ Behavioural & social sciences  ☒ Ecological, evolutionary & environmental sciences

For a reference copy of the document with all sections, see nature.com/documents/nr-reporting-summary-flat.pdf

# Ecological, evolutionary & environmental sciences study design

All studies must disclose on these points even when the disclosure is negative.

| | |
|---|---|
| Study description | Symbiotic deep-sea mussles of the genus Bathymodiolus and Gigantidas were analyzed to study the association with intra-nuclear bacteria |
| Research sample | Mussels were collected with remotely operated vehicles during two research cruises. Onboard, the mussels' gills were dissected, preserved, and stored for further processing. The symbiont housing organ (gill) was subjected to metagenome and metatrancriptome sequencing, metaproteomics and FISH analyses. |
| Sampling strategy | Mussels were sampled with a net from their natural habitat. Sampling of individuals depends on fieldwork conditions. Sampling sizes are sufficient for analyses performed in study |
| Data collection | none |
| Timing and spatial scale | Gigantidas childressi mussels were collected with the ROV Hercules during the RV Meteor Nautilus NA-58 cruise to the Gulf of Mexico in May 2015 at the Mississippi Canyon site (MC853, 28º07' N; -089º08' W) and the Green Canyon site (GC234, 27º45' N; -091º13' W) at water depths of 1,070 and 540 m, respectively. B. puteoserpentis mussels were collected with the ROV MARUM-QUEST during the Meteor M126 cruise to the Mid-Atlantic Ridge in April 2016 from the Logatchev vent field (Irina-II smoker, 14º45' N; -044º59' W) at a water depth of 3,036 m. |
| Data exclusions | no data were excluded |
| Reproducibility | Using the deposited raw sequencing, proteomic and imaging data, the data analyses that were performed in this study can be easily and repeatedly reproduced. |
| Randomization | Not relevant. To study the intra-nuclear association, samples were screened for the presence of the intra-nuclear parasite. |
| Blinding | Blinding was not performed because it was not relevant to this study. This study was an exploratory survey without a priori expectations that would influence the analyses. |

Did the study involve field work?  ☒ Yes  ☐ No

## Field work, collection and transport

| | |
|---|---|
| Field conditions | Deep-sea sampling at hydrothermal vents. Temperatures of mussel occurences were usually between 4 and 10 °C (for those where measurement was available). |
| Location | Gigantidas childressi mussels were collected with the remotely operated vehicle (ROV) Hercules during the RV Meteor Nautilus NA-58 cruise to the Gulf of Mexico in May 2015 at the Mississippi Canyon site (MC853, 28º07' N; -089º08' W) and the Green Canyon site (GC234, 27º45' N; -091º13' W) at water depths of 1,070 and 540 m, respectively. B. puteoserpentis mussels were collected with the ROV MARUM-QUEST during the Meteor M126 cruise to the Mid-Atlantic Ridge in April 2016 from the Logatchev vent field (Irina-II smoker, 14º45' N; -044º59' W) at a water depth of 3,036 m. |
| Access & import/export | Material used in this study were collected during German and US research cruises. Animals were collected from the deep/sea floor using remote operated vehicles. Bathymodiolus mussels are non-commercial and are not subjected to CITES or any other international regulations. Import permissions into Germany were granted by German authorities, where necessary. |
| Disturbance | All sampling adhered to the InterRigdeCode of conduct of work at hydrothermal vents (https://www.interridge.org/irstatement) |

# Reporting for specific materials, systems and methods

We require information from authors about some types of materials, experimental systems and methods used in many studies. Here, indicate whether each material, system or method listed is relevant to your study. If you are not sure if a list item applies to your research, read the appropriate section before selecting a response.

## Materials & experimental systems

| n/a | Involved in the study |
|---|---|
| ☒ | Antibodies |
| ☒ | Eukaryotic cell lines |
| ☒ | Palaeontology and archaeology |
| ☐ ☒ | Animals and other organisms |
| ☒ | Clinical data |
| ☒ | Dual use research of concern |
| ☒ | Plants |

## Methods

| n/a | Involved in the study |
|---|---|
| ☒ | ChIP-seq |
| ☒ | Flow cytometry |
| ☒ | MRI-based neuroimaging |

# Animals and other research organisms

Policy information about studies involving animals; ARRIVE guidelines recommended for reporting animal research, and Sex and Gender in Research

| | |
|---|---|
| Laboratory animals | No laboratory animals were used in this study. |
| Wild animals | Mussels of the species Gigantidas childressi and Bathymodiolus puteoserpentis were collected from hydrothermal vent and cold seep mussel fields with remotely operated vehicles operated from board of research vessels using nets. Mussels were transported in ambient water (4°C - 8°C) in a temperature isolated container to the surface where they were dissected and the tissue preserved for different experiments. |
| Reporting on sex | Sex was not considered for this study. |
| Field-collected samples | Adult mussels of the species Gigantidas childressi and Bathymodiolus puteoserpentis, of unknown age, with a shell length of 8-10cm were collected from hydrothermal vent and cold seep mussel fields with remotely operated vehicles operated from board of research vessels, using nets. Mussels were transferred into a temperature-isolated box full of ambient seawater, that kept them at ambient temperature and in the dark. They were transported to the surface where they were immediately dissected and the tissue preserved for different experiments. |
| Ethics oversight | Work on these mussels is not subjected to a approval by an ethics committee |

Note that full information on the approval of the study protocol must also be provided in the manuscript.

# Plants

| | |
|---|---|
| Seed stocks | No plants were used in this study |
| Novel plant genotypes | No plants were used in this study |
| Authentication | No plants were used in this study |

