## [Peer Review File · Nature Microbiology]

Peer Review Information

Journal: Nature Microbiology

Manuscript Title: An intranuclear bacterial parasite of deep-sea mussels expresses apoptosis inhibitors acquired from its host

Corresponding author name(s): Professor Nicole Dubilier

Reviewer Comments & Decisions:Decision Letter, initial version:

Message: 19th March 2024

Dear Nicole,

Thank you for your patience while your manuscript "An intranuclear bacterial parasite of deep-sea mussels expresses apoptosis inhibitors acquired from its host" was under peer-review at Nature Microbiology. It has now been seen by 3 referees, whose expertise and comments you will find at the of this email. You will see from their comments below that while they find your work of interest, some important points are raised. We are very interested in the possibility of publishing your study in Nature Microbiology, but would like to consider your response to these concerns in the form of a revised manuscript before we make a final decision on publication.

In particular, you will see that all referees have some questions or comments that require clarification, additional details or additional analysis to address. The rest referees' reports are clear and the remaining issues should be straightforward to address.

If you have not done so already please begin to revise your manuscript so that it conforms to our Article format instructions at <http://www.nature.com/nmicrobiol/info/final-submission/>

The usual length limit for a Nature Microbiology Article is six display items (figures or tables) and 3,000 words. We have some flexibility, and can allow a revised manuscript at 3,500 words, but please consider this a firm upper limit. There is a trade-off of ~250 words per display item, so if you need more space, you could move a Figure or Table to Supplementary Information.

Some reduction could be achieved by focusing any introductory material and moving it to the start of your opening 'bold' paragraph, whose function is to outline the background to your work, describe in a sentence your new observations, and explain your main conclusions. The discussion should also be limited. Methods should be described in a separate section following the discussion, we do not place a word limit on Methods.

Nature Microbiology titles should give a sense of the main new findings of a manuscript, and should not contain punctuation. Please keep in mind that we strongly discourage active verbs in titles, and that they should ideally fit within 90 characters each (including spaces).

We strongly support public availability of data. Please place the data used in your paper into a public data repository, if one exists, or alternatively, present the data as Source

3Data or Supplementary Information. If data can only be shared on request, please explain why in your Data Availability Statement, and also in the correspondence with your editor. For some data types, deposition in a public repository is mandatory - more information on our data deposition policies and available repositories can be found at <https://www.nature.com/nature-research/editorial-policies/reporting-standards#availability-of-data>.

Please include a data availability statement as a separate section after Methods but before references, under the heading "Data Availability". This section should inform readers about the availability of the data used to support the conclusions of your study. This information includes accession codes to public repositories (data banks for protein, DNA or RNA sequences, microarray, proteomics data etc...), references to source data published alongside the paper, unique identifiers such as URLs to data repository entries, or data set DOIs, and any other statement about data availability. At a minimum, you should include the following statement: "The data that support the findings of this study are available from the corresponding author upon request", mentioning any restrictions on availability. If DOIs are provided, we also strongly encourage including these in the Reference list (authors, title, publisher (repository name), identifier, year). For more guidance on how to write this section please see: <http://www.nature.com/authors/policies/data/data-availability-statements-data-citations.pdf>

To improve the accessibility of your paper to readers from other research areas, please pay particular attention to the wording of the paper's opening bold paragraph, which serves both as an introduction and as a brief, non-technical summary in about 150 words. If, however, you require one or two extra sentences to explain your work clearly, please include them even if the paragraph is over-length as a result. The opening paragraph should not contain references. Because scientists from other sub-disciplines will be interested in your results and their implications, it is important to explain essential but specialised terms concisely. We suggest you show your summary paragraph to colleagues in other fields to uncover any problematic concepts.

If your paper is accepted for publication, we will edit your display items electronically so they conform to our house style and will reproduce clearly in print. If necessary, we will re-size figures to fit single or double column width. If your figures contain several parts, the parts should form a neat rectangle when assembled. Choosing the right electronic format at this stage will speed up the processing of your paper and give the best possible results in print. We would like the figures to be supplied as vector files - EPS, PDF, AI or postscript (PS) file formats (not raster or bitmap files), preferably generated with vector-graphics software (Adobe Illustrator for example). Please try to ensure that all figures are non-flattened and fully editable. All images should be at least 300 dpi resolution (when figures are scaled to approximately the size that they are to be printed at) and in RGB colour format. Please do not submit Jpeg or flattened TIFF files. Please see also 'Guidelines for Electronic Submission of Figures' at the end of this letter for further detail.

Figure legends must provide a brief description of the figure and the symbols used, within 350 words, including definitions of any error bars employed in the figures.

When submitting the revised version of your manuscript, please pay close attention to our [href="https://www.nature.com/nature-research/editorial-policies/image-integrity">Digital Image Integrity Guidelines](https://www.nature.com/nature-research/editorial-policies/image-integrity). and to the following points below:

Please include a statement before the acknowledgements naming the author to whom correspondence and requests for materials should be addressed.

Finally, we require authors to include a statement of their individual contributions to the paper -- such as experimental work, project planning, data analysis, etc. -- immediately after the acknowledgements. The statement should be short, and refer to authors by their initials. For details please see the Authorship section of our joint Editorial policies at http://www.nature.com/authors/editorial_policies/authorship.html

- * include a point-by-point response to any editorial suggestions and to our referees. Please include your response to the editorial suggestions in your cover letter, and please upload your response to the referees as a separate document.

- * ensure it complies with our format requirements for Letters as set out in our guide to authors at www.nature.com/nmicrobiol/info/gta/

- * state in a cover note the length of the text, methods and legends; the number of references; number and estimated final size of figures and tables

- * resubmit electronically if possible using the link below to access your home page:

*This url links to your confidential homepage and associated information about manuscripts you may have submitted or be reviewing for us. If you wish to forward this e-mail to co-authors, please delete this link to your homepage first.

Please ensure that all correspondence is marked with your Nature Microbiology reference number in the subject line.

Nature Microbiology is committed to improving transparency in authorship. As part of our efforts in this direction, we are now requesting that all authors identified as 'corresponding author' on published papers create and link their Open Researcher and Contributor Identifier (ORCID) with their account on the Manuscript Tracking System (MTS), prior to acceptance. This applies to primary research papers only. ORCID helps the scientific community achieve unambiguous attribution of all scholarly contributions. You can create and link your ORCID from the home page of the MTS by clicking on 'Modify my Springer

Nature account'. For more information please visit please visit www.springernature.com/orcid.

We hope to receive your revised paper within four weeks. If you cannot send it within this time, please let us know.

Yours sincerely,

Reviewer Expertise:

Referee #1: symbiosis, nuclear bacteria, phylogenetics
Referee #2: symbiosis, intracellular bacteria, omics
Referee #3: marine symbioses, omics

Reviewers Comments:

Reviewer #1 (Remarks to the Author):

The manuscript from Porras and colleagues deals with a fascinating system and an interesting topic. It is well thought-out, clearly organized, and scientifically sound. We have a few suggestions for improvements, and they should be easy to implement as they are mostly about the text, and some context.

Major suggestions:

The comparative discussion and reference choices on some topics should be improved.

The main examples are:

Comments and citations on other nuclear bacteria. The authors acknowledge that most of the known cases are in protists, but even so, they only cite a 9-year old review that doesn't include the most recent and relevant obvious comparisons (bacteria that do seem to consume the host's chromatin, that also have viral particles in their cytoplasm, that leave the nucleus in various ways, etc.). Also, around 438 bacteria in protists are discussed, but the authors really should mention that the vast majority of those cases that have been described are bacteria that seem to be quite benign. They don't seem to really "do" anything, they are just commensals. They are quite different from this clearly deleterious bacterium. This reminds us more of *Dependentiae* bacteria.

The discussion on horizontal gene transfer is a bit superficial and the authors should cite maybe more recent papers. Their HGT example is very interesting and looks solid, so this is just a discussion/context comment. For example, around Line 393 they discuss why people think bacteria don't get eukaryotic genes and cite a paper from over 10 years ago. However, their argument is not a great one, and is not in that paper. DNA is not inaccessible because it is in the nucleus, because donor DNA for HGT is not likely from a live cell – it's more likely DNA from a dead cell, or it's being moved by a virus etc., but either way the nucleus is hardly a factor. The better argument is that bacteria just don't have much use for genes from eukaryotes as bacteria capitalize more on metabolic variability than eukaryotes. This is what the paper they cite suggested, and another similar review just came out that makes this case in the opposite direction as well

(Keeling 2024 Nat. Rev. Genet.).

Same for caspases and apoptosis, maybe update with a few more references.

The tree in Figure 1b has a few issues. First, it should be re-run with a more refined software than FastTree 2, which is only supposed to be used for quick, preliminary analyses. The later tree is performed at a much higher standard, so why not this one? Also, the figure legend mentions Posterior Probability values, but FastTree is an approximate ML software, and the authors themselves mention bootstrapping in the caption, so which is it? Did they do Bayesian analyses? It seems sloppy and makes us wonder what the analysis actually was, so it is suggested to make a better tree and describe it properly. Lastly, and most significantly, the tree does not really show what the authors say it does. Perhaps this will change with a better analysis, but the two new taxa fall in a position with no support - every node adjacent to them is unsupported, so you can't really say where they go. This is a problem with no other information, since theoretically these could fall in a completely different group for all you can tell from this tree. We assume they did other trees that showed the bacteria do fall in this group, but at face value this tree says almost nothing about the position of these two taxa.

In figure 2 the membrane labeling is odd, or rather lacks explanation. The authors have labeled the membranes with two different labels each. What they mean could be inferred (when it's in the nucleus vs. when it's invading the cell, probably?), but this should be explained.

TYPOS AND MARGINAL COMMENTS

Two small comments on nomenclature: (I) in bacteria, all taxa names (not just genus and species) are supposed to be italicized; (II) "Candidatus Genera" require quotations marks around their name

Line 40: typo, "ubiquitous"

Line 93: this is a fascinating result, possibly not emphasized enough in the manuscript. Nevertheless, we suggest drawing lines around the borders of cells in the fluorescence micrographs in Figure 1a (at first glance, it is not apparent that the infected cells have no other symbionts, especially in the "early" and "late" photos). Also, maybe the authors might add here an approximate number of how many infected cells they have inspected overall, that never had cytoplasmic symbionts?

Line 112: since this is the first occurrence, please spell out the new names in full

Lines 196-199: interesting speculation, but then the sugars from the degradation of mucin should be transported back through the host membrane, nuclear envelope, and bacterial membranes, correct?

Line 201: typo, "it nutrition"

Line 343: typo, "capase-2"

Line 355: where it says "encounters" do you mean "counters"

Lines 370-371: the sentence is a bit confusing, should be made more precise

Line 406: typo, "Crassotrea"

Figure 4: the colours for Endozoicomonas and ascidians are almost indistinguishable in print

Lines 437-438: is there a citation for this, or is this a result of the authors' analyses?

Line 439: this is correct, but elsewhere in the paper the authors state that apoptosis is an exclusively metazoan feature, so that should be qualified

Lines 440-445: one could make the counterpoint that vertically inherited nuclear bacteria

are under similar "timing" selection because they need to divide before their host does (but not too much). As another hypothesis following directly from what the authors are saying, what if an IAP-lacking ancestor was already intranuclear, and simply did not divide enough to trigger apoptosis? When, at a later point, it somehow acquired IAPs, this would allow it to become highly proliferative without killing the host too quickly, once it could stop the host's apoptosis

Line 479: typo, "4%paraformaldehyde"

Lines 490-491: "16S rRNA oligonucleotide probes" makes it sound like the probe itself is RNA, while it is likely DNA (binding to RNA in the cell), correct?

Line 493: typo, ")")"

Line 591: typo, "in the next section below"

Line 666: typo, "leads"

Extended data Fig. 2

Reviewer #2 (Remarks to the Author):

This manuscript reports on the genomic, transcriptomic, and proteomic analysis of bacteria infecting the nuclei of gill cells of deep sea mussels. Our knowledge about bacteria living within the nucleus of their host cells is sparse. The manuscript provides exciting insights into this unusual lifestyle in an experimentally challenging biological system. The two main findings are the (unexpected) physiology of these bacteria and the discovery of putative apoptosis inhibitors likely co-opted from their animal hosts.

My comments focus on these two main findings:

1. How are high quality genomes defined here (line 108) - according to the MIMAG standard? Supp. Table 2 suggests that both are incomplete genome sequences with >10 contigs - or high-quality draft MAGs. This should be mentioned, and the reader should be referred to Supp. Table 2 for the assembly details.
2. Metabolism of *Endonucleobacter* / Supplementary Tables 3-8: How does this information help to understand that *Endonucleobacter* "consume sugars, lipids and amino acids from their host"? It may be largely unclear to many readers how the authors arrived at conclusions such as "the host expressed genes for import of sugars, amino acids and the synthesis of lipid droplets". For this purpose, it might be helpful to assign the genes listed in the supp. tables to cellular functions or metabolic pathways. This is shown in Figure 3, but it's somewhat difficult to recognize this in the original data.

I applaud the authors for the successfully performing dual RNA-Seq experiments in this experimentally challenging system, but I still have a few questions about the data analysis:

3. The analysis of the RNA-Seq data is not described in the methods section of the main manuscript, and I haven't found any details about this in the supplementary material. For clarity, all methods should be included in the main manuscript.
4. Reproducibility: How well did the expression patterns of the LMD replicates agree? The difference between replicates appears to be substantial in some of the supplementary tables.

5. Gene expression analysis and Figure 3: Which gene expression changes were considered to be significantly different?
Does the change in host gene expression at the early time point refer to uninfected cells?
Why is bacterial gene expression not expressed as a fold change from the previous time point?
6. The authors suggest that *Endonucleobacter* secretes the chitinase ChiA via the T3SS. Are T3SS genes differentially expressed at any stage of the infection?
7. Were homologs of nucleotide transporters of the TLC family - known from intracellular bacteria - found in the *Endonucleobacter* genomes?
8. Similarity searches for IAPs: The discovery of several predicted apoptosis inhibitors in the *Endonucleobacter* genomes is interesting. It remains unclear, however, which conserved IAP sequences were searched for. It would be good to include this information and to provide the corresponding hmm profiles as supplementary data.
9. Seven predicted IAPs are listed for the *E. childressi* genome in the supplementary data, but I haven't found any details about the IAPs found in *E. puteoserpentis*.
10. Inhibition of programmed cell death is a common strategy among intracellular bacteria that infect animals, see e.g. <https://www.ncbi.nlm.nih.gov/pmc/articles/PMC6800630/>
It would be useful to discuss (very briefly) the *Endonucleobacter* IAPs in the context of known effector proteins with anti-apoptotic activity, perhaps by extending the relevant paragraph in the conclusions section.
11. As the predicted IAPs contain a signal peptide for secretion via the Sec system, would this imply that they act on caspases located within the nucleus as opposed to cytoplasmic caspases?
12. IAP phylogeny: How comprehensive is this dataset in terms of bacterial sequences? As the Endozoicomonadaceae IAPs were identified using the hmm profiles generated in this study, is it possible that there are additional IAPs present in other bacteria? If a more comprehensive bacterial genome data set was analysed, would this yield hits in other bacterial genomes, and if so would this change the phylogenetic inferences?
13. The statement in the abstract that IAPs were "repeatedly [acquired] through horizontal gene transfer (HGT) from their hosts in convergent acquisition" is not well-supported by the phylogenetic analysis. The tree does suggest repeated acquisition events, but rather than through independent events, the IAPs appear to have been acquired by the ancestor of the two *Endonucleobacter* species, followed by subsequent co-diversification and differential gene loss.
14. The observation of virus particles in *Endonucleobacter* cells is fascinating. However, as icosahedral capsids are not restricted to dsDNA viruses, the speculation that they might represent the host virus OshV-1 seems far-fetched.
Is there any evidence for phages or pro-phages in the sequence assemblies?
15. Data availability and traceability of genome annotation: The *Endonucleobacter* genome sequences at Genbank/ENA/DDBJ should include the complete annotation used in the present study. Supplementary tables should include the final locus tags used for submission of the annotated genome sequence.

Similarly, the *G. childressi* de novo transcriptome including the annotation must be submitted and available. The gene names and generic locus tags (trinity something) used in the manuscript for host genes are largely useless to readers interested in specific gene/protein sequences.

Are all original sequence data sets available in SRA?

Minor points:

16. Figure 1a is not referred to in the manuscript; should be included in the first paragraph of the results and discussion section.

17. Figure 1a: Which *Endonucleobacter* species is shown? *E. puteoserpentis*?

18. Figure 2: Gene expression data from which infection stage? From the “bulk transcriptomics” experiment?

19. L. 139, 143, “these were expressed 139 in *Ca. E. childressi*.”: Are the respective genes or proteins expressed? At which infection stage?

20. Supplementary Tables 3 and 4 appear to be identical.

21. Supplementary Table 2 and 5: *E. puteoserpentis* is named *E. bathymodioli* in this table.

22. Figure 4: *E. puteoserpentis* is named *E. bathymodioli* in this figure.

23. Figure 4 legend: Typo, IQTREET should read IQ-TREE

24. Supplementary notes 3 and 4 are not referred to in the main text.

Reviewer #3 (Remarks to the Author):

The introduction gives a relevant background to the bacterial clade of *Ca. Endonucleobacter*, which infects the nucleus of mussel cells. Clear research questions are stated that are related to cellular and molecular process that could support the replication and survival of the intra-nuclear *Ca. Endonucleobacter* cells.

The Material and Methods are described in a manner to reproduce the work and are performed to high technical standards.

The Results and Discussion section initially describes the localisation of *Ca.*

Endonucleobacter in two mussels species, which show an interesting exclusion pattern with other symbiotic bacteria, such as sulfur- and / or methane-oxidizing bacteria. While I don't expect that this study can fully explain these exclusion patterns, I wonder if this could be picked up later in the Discussion or Conclusion given that sulfur- and / or methane-oxidizing bacteria would provide much energy to mussel cells, something that the immense reproduction of *Ca. Endonucleobacter* would certainly benefit from.

Genome reconstruction further defined that the *Ca. Endonucleobacter* in two mussels species represent two distinct species, which have much smaller genomes compared to related *Endozoicomonas* species. A loss of amino acid synthesis pathways is mentioned

and it would have been of interest to understand what other genetic feature underpin the apparent genome reduction.

A careful and state-of-the-art genomic and transcriptomic analysis indicate that *Ca. Endonucleobacter* mostly utilises sugar, peptides and lipids, and likely not much nucleic acids. This is also consistent with the morphological and expression phenotypes observed for the host. I note though that the transcriptomic and proteomic data in Table S3, S4 and S7, which the section about gene expression refers to (i.e. L121ff), only shows 30-50 genes/proteins, which I assume is not all that is being detected. I could not find any headings or explanations for the Supplementary Tables, which made understanding them quite challenging. Please provide more context for them.

The *Ca. Endonucleobacter* genomes further contain a number of bona fide IAPs and their expression patterns indicate a key role in facilitating the survival of the bacterium by blocking the apoptosis of the host. Evidence is presented that IAPs are acquired via multiple HGT as the authors identified eight clades where bacterial and non-bacterial sequences are interspersed. And finally, evidence is provided that *Ca. Endonucleobacter* is infected by a virus that could mediate such a HGT. The scenario described in the manuscript about how IAPs function and evolved are not overly speculative and very interesting.

The conclusion provides interesting connections to other work on intra-nuclear/cellular symbionts, Endozoicomonadaceae associations and eukaryote-to-prokaryote HGT, and this caps off a well-written and well-executed study that makes a significant contribution to our understanding of bacteria-host interactions.

Minor comments:

Line 19: Replace "digesting" by "digested".

Line 40: Replace "ubiquotous" with "ubiquitous".

Line 93: Provide cross-link to Extended Data Fig. 1 here.

L162: If the three *Hahella* genomes were used to root the tree, why is there then another root on the left side of the tree? This would indicate that some other sequence(s) was used to place the root between the two major clusters.

Line 184: Define "T3SS".

Line 494: Fix ")))"

L679: Please explain what you mean by "pseudoalign".

Author Rebuttal to Initial comments

Response to reviewers

We are very grateful to the reviewers for their helpful comments and constructive feedback. We have revised our manuscript and explain in our point-by-point replies below how we have addressed the reviewers' comments. These are in italics, while our replies are in plain font.

Reviewer Expertise:

Referee #1: symbiosis, nuclear bacteria, phylogenetics

Referee #2: symbiosis, intracellular bacteria, omics

Referee #3: marine symbioses, omics

Reviewers Comments:

Reviewer #1 (Remarks to the Author):

The manuscript from Porras and colleagues deals with a fascinating system and an interesting topic. It is well thought-out, clearly organized, and scientifically sound.

Thank you for your kind comments.

We have a few suggestions for improvements, and they should be easy to implement as they are mostly about the text, and some context.

Major suggestions:

The comparative discussion and reference choices on some topics should be improved. The main examples are:

Comments and citations on other nuclear bacteria. The authors acknowledge that most of the known cases are in protists, but even so, they only cite a 9-year old review that doesn't include the most recent and relevant obvious comparisons (bacteria that do seem to consume the host's chromatin, that also have viral particles in their cytoplasm, that leave the nucleus in various ways, etc.).

We thank the reviewer for their suggestion, and have added two more recent reviews (Husnik et al. 2021; Schrollhammer & Potekhin 2020).

Also, around 438 bacteria in protists are discussed, but the authors really should mention that the vast majority of those cases that have been described are bacteria that seem to be quite benign. They don't seem to really "do" anything, they are just commensals. They are quite different from this clearly deleterious bacterium. This reminds us more of Dependientiae bacteria.

We agree and clarified the differences in lifestyles between "Ca. Endonucleobacter" and bacteria found in protists – line 448 – 451.

“Moreover, if a bacterium that lives in a unicellular host is passed on to both daughter cells, it could lead a sheltered, intranuclear lifestyle as long as the bacterium does not have major negative effects on its host's fitness. Indeed, some of the intranuclear bacteria that colonize protists can be benign (Bella et al., 2016; Schulz et al., 2014).

The discussion on horizontal gene transfer is a bit superficial and the authors should cite maybe more recent papers.

We agree and have now revised our discussion (see reply to next comment) and cited more recent papers (Haimlich et al., 2024; Keeling & Palmer, 2008; Li et al., 2024; Zilber-Rosenberg & Rosenberg, 2021)).

Their HGT example is very interesting and looks solid, so this is just a discussion/context comment. For example, around Line 393 they discuss why people think bacteria don't get eukaryotic genes and cite a paper from over 10 years ago. However, their argument is not a great one, and is not in that paper. DNA is not inaccessible because it is in the nucleus, because donor DNA for HGT is not likely from a live cell – it's more likely DNA from a dead cell, or it's being moved by a virus etc., but either way the nucleus is hardly a factor. The better argument is that bacteria just don't have much use for genes from eukaryotes as bacteria capitalize more on metabolic variability than eukaryotes. This is what the paper they cite suggested, and another similar review just came out that makes this case in the opposite direction as well (Keeling 2024 Nat. Rev. Genet.).

We agree with the reviewer's comments, have removed the part about the inaccessibility of the DNA and have revised accordingly – line 456 - 463

” Our study adds to the small, but growing body of evidence for HGT from eukaryotes to bacteria (Haimlich et al., 2024; Keeling & Palmer, 2008; Li et al., 2024; Zilber-Rosenberg & Rosenberg, 2021). HGT from eukaryotes to prokaryotes is assumed to be disfavored for several reasons including the lack of eukaryotic metabolic versatility compared to bacteria, and the presence of eukaryotic introns as barriers to genetic transfer of genes to prokaryotes (Keeling & Palmer, 2008). Factors that favor HGT from eukaryotes to prokaryotes include

13intimate interactions between these, as is the case for "Ca. Endonucleobacter" and its mussel hosts, and strong selection for the acquisition of eukaryotic genes by prokaryotes (Keeling, 2024).)"

Same for caspases and apoptosis, maybe update with a few more references.

We have updated the text with more recent references (Green, 2022; Lalaoui & Vaux, 2018; Ramirez & Salvesen, 2018).

The tree in Figure 1b has a few issues. First, it should be re-run with a more refined software than FastTree 2, which is only supposed to be used for quick, preliminary analyses. The later tree is performed at a much higher standard, so why not this one? Also, the figure legend mentions Posterior Probability values, but FastTree is an approximate ML software, and the authors themselves mention bootstrapping in the caption, so which is it? Did they do Bayesian analyses? It seems sloppy and makes us wonder what the analysis actually was, so it is suggested to make a better tree and describe it properly. Lastly, and most significantly, the tree does not really show what the authors say it does. Perhaps this will change with a better analysis, but the two new taxa fall in a position with no support - every node adjacent to them is unsupported, so you can't really say where they go. This is a problem with no other information, since theoretically these could fall in a completely different group for all you can tell from this tree. We assume they did other trees that showed the bacteria do fall in this group, but at face value this tree says almost nothing about the position of these two taxa.

We agree and have rerun our analysis with a newer, more comprehensive dataset and a more refined pipeline. Genes were identified and aligned with the GToTree pipeline, the tree calculated with IQTREE, and branch support (1000 replicates) calculated with both SH-aLRT and UFBoot (we added this information to the legend of Fig. 1). The revised tree shows that "Ca. Endonucleobacter" forms a well-supported clade, with the genus *Endozoicomonas* as its sister clade based on 100% support (Fig. 1b).

In figure 2 the membrane labeling is odd, or rather lacks explanation. The authors have labeled the membranes with two different labels each. What they mean could be inferred (when it's in the nucleus vs. when it's invading the cell, probably?), but this should be explained.

We have revised the labeling in Figure 2, and now labeled and named the inner and outer nuclear membrane in the figure.

TYPOS AND MARGINAL COMMENTS

Line 93: this is a fascinating result, possibly not emphasized enough in the manuscript. Nevertheless, we suggest drawing lines around the borders of cells in the fluorescence micrographs in Figure 1a (at first glance, it is not apparent that the infected cells have no other symbionts, especially in the “early” and “late” photos). Also, maybe the authors might add here an approximate number of how many infected cells they have inspected overall, that never had cytoplasmic symbionts?

We thank the reviewer for their suggestion and have drawn dotted lines in Figure 1A that indicate the cell border of the neighboring, symbiont-containing bacteriocytes and explain the dotted lines in the figure legend.

On the number of cells we observed, we revised the text to – line 88-91

“Our FISH analyses of thousands of cells from at least ten mussel individuals collected over several decades revealed that in both mussel species, the parasite never infected cells with symbiotic bacteria (**Extended Data Fig. 1**).”

Extended Figure 1 shows overviews of whole gill filaments and the distribution of the parasite across thousands of host cells. The exclusion of “*Ca. Endonucleobacter*” from symbiont-containing cells is particularly apparent in *Gigantidas childressi*, where “*Ca. Endonucleobacter*” only colonized host cells at the ciliated edges of the gill, which are always free of symbionts.

Lines 196-199: interesting speculation, but then the sugars from the degradation of mucin should be transported back through the host membrane, nuclear envelope, and bacterial membranes, correct?

Correct. We have expanded on this hypothesis in lines 199-203:

“Mucin-derived sugars could be taken up by the mussel through its *SWEET* importer and degraded in the cytoplasm to GlcNAc by the chitinase *CTBS*, as both genes were upregulated by the host in early and mid infection stages. The resulting cytoplasmic GlcNAc could then diffuse into the nucleus, and be taken up by “*Ca. Endonucleobacter*” via its phosphotransferase system *PTS*.”

Lines 370-371: the sentence is a bit confusing, should be made more precise

We agree and have revised to – line 378-380:

“IAPs in all three “*Ca. Endonucleobacter*” clades were most closely related to those of their bathymodioline hosts, as well as other mollusks (Fig. 4).”

Figure 4: the colours for Endozoicomonas and ascidians are almost indistinguishable in print

Thank you for pointing this out, we adjusted the colors.

Lines 437-438: is there a citation for this, or is this a result of the authors' analyses?

This is based on our own analyses, and has been clarified in the text – line 445-447:

“For example, the intranuclear bacteria that colonize protists do not have IAPs, based on our database queries.”

Line 439: this is correct, but elsewhere in the paper the authors state that apoptosis is an exclusively metazoan feature, so that should be qualified

Good point, thank you. We have revised our text to clarify that caspase-mediated apoptosis is an exclusively metazoan feature – line 393-396:

“Given that caspase-mediated apoptosis is specific to animals, and IAPs are only known from animals and some viruses that acquired IAPs horizontally from their invertebrate hosts, IAPs in bacteria are likely not ancestral, but were rather acquired through HGT from animals or viruses.”

Lines 440-445: one could make the counterpoint that vertically inherited nuclear bacteria are under similar “timing” selection because they need to divide before their host does (but not too much). As another hypothesis following directly from what the authors are saying, what if an IAP-lacking ancestor was already intranuclear, and simply did not divide enough to trigger apoptosis? When, at a later point, it somehow acquired IAPs, this would allow it to become highly proliferative without killing the host too quickly, once it could stop the host's apoptosis.

Yes, we agree with the reviewer and have made this point clear in our revision - line 428 – 433:

“If colonization of the nucleus by the pathogen occurred first before acquiring IAPs, then division rates of ancestral “Ca. Endonucleobacter” would have had to have been low enough to not induce apoptosis. Alternatively, if “Ca. Endonucleobacter” acquired IAPs before its intranuclear lifestyle, this would have required at least two steps: i) intimate contact with the host's DNA or mRNA or that of a virus of the host, and ii) a certain frequency of this contact.”

Lines 490-491: “16S rRNA oligonucleotide probes” makes it sound like the probe itself is RNA, while it is likely DNA (binding to RNA in the cell), correct?

Correct, thank you. We now revised to:
“oligonucleotide probes targeting 16S rRNA...”

TYPOS

Two small comments on nomenclature: (I) in bacteria, all taxa names (not just genus and species) are supposed to be italicized; (II) “Candidatus Genera” require quotations marks around their name

Thank you for pointing this out, we have revised accordingly throughout our text.

Line 40: typo, “ubiquotous”

Line 112: since this is the first occurrence, please spell out the new names in full

Line 201: typo, “it nutrition”

Line 343: typo, “capase-2”

Line 355: where it says “encounters” do you mean “counters”

Line 406: typo, “Crassotrea”

Line 479: typo, “4%paraformaldehyde”

Line 493: typo, “)”)”

Line 591: typo, “in the next section below”

Line 666: typo, “leads”

Extended data Fig. 2

Thank you for pointing out these typos, they have now all been corrected.

Reviewer #2 (Remarks to the Author):

This manuscript reports on the genomic, transcriptomic, and proteomic analysis of bacteria infecting the nuclei of gill cells of deep sea mussels. Our knowledge about bacteria living within the nucleus of their host cells is sparse. The manuscript provides exciting insights into this unusual lifestyle in an experimentally challenging biological system. The two main findings are the (unexpected) physiology of these bacteria and the discovery of putative apoptosis inhibitors likely co-opted from their animal hosts.

Thank you for your kind comments.

My comments focus on these two main findings:

1. How are high quality genomes defined here (line 108) - according to the MIMAG standard? Supp. Table 2 suggests that both are incomplete genome sequences with >10 contigs – or high-quality draft MAGs. This should be mentioned, and the reader should be referred to Supp. Table 2 for the assembly details.

17We agree and have added this information to Supplementary Table 2 and refer to the table in the main text – line 112 -116:

“Our analyses of high-quality draft metagenome-assembled genomes (Bowers et al., 2017) (Supplementary Information Table 2), assembled from both short- and long-read sequencing of *B. puteoserpentis* and *G. childressi* gill tissues, revealed that these two mussel species are infected by genetically distinct “Ca. Endonucleobacter” species, based on their average nucleotide identity of only 84.3%.”

2. Metabolism of Endonucleobacter / Supplementary Tables 3-8: How does this information help to understand that Endonucleobacter “consume sugars, lipids and amino acids from their host”? It may be largely unclear to many readers how the authors arrived at conclusions such as “the host expressed genes for import of sugars, amino acids and the synthesis of lipid droplets”. For this purpose, it might be helpful to assign the genes listed in the supp. tables to cellular functions or metabolic pathways. This is shown in Figure 3, but it’s somewhat difficult to recognize this in the original data.

We are grateful to the reviewer for their suggestion and have added the column “Cellular function” to the respective supplementary tables (Supplementary Tables 3,4,7 and 8), which form the basis for the reconstructions shown in Figures 2 and 3.

I applaud the authors for the successfully performing dual RNA-Seq experiments in this experimentally challenging system, but I still have a few questions about the data analysis:

3. The analysis of the RNA-Seq data is not described in the methods section of the main manuscript, and I haven’t found any details about this in the supplementary material. For clarity, all methods should be included in the main manuscript.

Thank you for pointing this out. We have added the analysis steps in the methods section of the main manuscript, line 651 -672 and 697 – 703.

4. Reproducibility: How well did the expression patterns of the LMD replicates agree? The difference between replicates appears to be substantial in some of the supplementary tables.

The reviewer is correct, we observed considerable variability between replicates. Most of this variability likely originated from the challenges inherent to LMD, as the transcriptomic readout came from samples with minute amounts of RNA that had gone through formalin fixation and the LMD processing procedure. This variability was more pronounced for the host

transcriptome, where transcript coverage, that is the percent of transcripts with at least one assigned read, was relatively low and ranged from 10 - 25% across infection stages.

We have added a NMDS plots of the transcriptomic replicates in our new Supplementary Figure 2. For the host, these analyses show that non-infected and early-infected samples were distinct from each other, and for the other two infection stages, at least two of the three replicates clustered most closely to each other. For the parasite, the differences between infection stages are more clear-cut and for infection stages, the three replicates clustered closely to each other and were distinct from other infection stages.

5. Gene expression analysis and Figure 3: Which gene expression changes were considered to be significantly different?

Why is bacterial gene expression not expressed as a fold change from the previous time point?

We agree, these are important questions that we considered in detail before analyzing our data. We decided to show changes in expression by comparing normalized expression between infection stages. We made this decision based on the following considerations:

- Differential expression analysis can be biased, as it can ignore highly expressed genes if they do not differ significantly in expression between stages.
- The dataset was generated from LMD samples that went through lengthy processing (including fixation with PFA), which limits the total amount of RNA available for analysis. Our approach allowed us to consider that genes can have 0 expression over multiple stages, and thus provides an overview of the complete spectrum of data points.:
- RNA-seq data is inherently compositional because the number of fragments sequenced is not proportional to the sample's total RNA content (Fernandes, 2014 <https://doi.org/10.1186/2049-2618-2-15>).
- To the question about bacterial gene expression:

For the host we generated transcriptomic profiles of non-infected cells as a baseline to compare the early infection stage with. However, for “*Ca. Endonucleobacter*”, we did not have a similar baseline, because they are not (yet) cultivable and we therefore cannot compare a ‘pre-infection’ transcriptome of these bacteria to the earliest infection stage. We therefore analyzed the transcriptomes of the three infection stages of “*Ca. Endonucleobacter*” using TPM normalization.

Does the change in host gene expression at the early time point refer to uninfected cells?

Yes, this is the change in expression between the non-infected cells and the early infected cells. We point this out in the legend of Figure 3 – line 298-300:

“Gene expression of *G. childressi* cells in the early stage of infection were compared to non-infected *G. childressi* cells.”

as well as in the methods section, line line 665-666.

6. *The authors suggest that Endonucleobacter secretes the chitinase ChiA via the T3SS. Are T3SS genes differentially expressed at any stage of the infection?*

Yes, we consistently observed upregulation of the T3SS genes from early to mid to late infection stages. We now provide this information in the new Supplementary Table 17.

7. *Were homologs of nucleotide transporters of the TLC family - known from intracellular bacteria - found in the Endonucleobacter genomes?*

No nucleotide transporters of the TLC family were found, and we have added this in the text – line 250 – 253.

“Both “*Ca. Endonucleobacter*” species lacked nucleotide importers of the TLC family, such as ADP/ATP translocases, known from intranuclear bacteria of protists like *Rickettsiales*, *Caedibacter caryophilus* and *Holospora* spp.. (Haferkamp et al., 2006; Schmitz-esser et al., 2004; Tjaden et al., 1999).

8. *Similarity searches for IAPs: The discovery of several predicted apoptosis inhibitors in the Endonucleobacter genomes is interesting. It remains unclear, however, which conserved IAP sequences were searched for. It would be good to include this information and to provide the corresponding hmm profiles as supplementary data.*

Thank you for pointing this out, we agree and have now provided a FASTA file with the sequences used to generate the HMM profile, together with the HMM profile itself, on Zenodo. Link here <https://zenodo.org/doi/10.5281/zenodo.11086255> and in the data availability section of the manuscript – line 746-755.

9. *Seven predicted IAPs are listed for the E. childressi genome in the supplementary data, but I haven't found any details about the IAPs found in E. puteoserpentis.*

Good point, we have now added this information to Extended Data Figure 5.

10. *Inhibition of programmed cell death is a common strategy among intracellular bacteria that infect animals, see e.g. <https://www.ncbi.nlm.nih.gov/pmc/articles/PMC6800630/>*

It would be useful to discuss (very briefly) the Endonucleobacter IAPs in the context of known effector proteins with anti-apoptotic activity, perhaps by extending the relevant paragraph in the conclusions section.

We agree and have added a review by Behar & Briken (2019) that describes other, IAP-independent, mechanisms of bacteria interfering with apoptosis in line 445.

11. *As the predicted IAPs contain a signal peptide for secretion via the Sec system, would this imply that they act on caspases located within the nucleus as opposed to cytoplasmic caspases?*

Indeed, secretion via the Sec system would translocate the bacterial IAPs into the nucleus. In model animal organisms like *C. elegans* and mice, initiator caspases like Caspase-2 were shown to possess nuclear import motifs and were localized in the nucleus (see e.g. doi:10.3390/cells9051259). We have now added this information in the text – line 357 – 360 :

“While secretion of bacterial IAPs into the nucleus could appear counterintuitive, caspase-2 has been shown to localize to, and induce apoptosis from the nucleus, for example as a reaction to DNA damage (Brown-Suedel & Bouchier-Hayes, 2020; Paroni et al., 2002).”

12. *IAP phylogeny: How comprehensive is this dataset in terms of bacterial sequences? As the Endozoicomonadaceae IAPs were identified using the hmm profiles generated in this study, is it possible that there are additional IAPs present in other bacteria? If a more comprehensive bacterial genome data set was analysed, would this yield hits in other bacterial genomes, and if so would this change the phylogenetic inferences?*

We thank the reviewer for their questions. We used the HMM profile we generated for IAP screening to query the UniProt database for bacterial entries. We did find a few hits from bacteria from other groups besides *Endozoicomonadaceae* and added this information in the manuscript – line 441 – 444 - and the details in Supplementary Table 18. We prefer not to add these sequences to our tree because their origin is not clear.

“Beyond *Endozoicomonadaceae*, our database queries recovered a few bacterial IAP sequences in samples collected from soils, the deep sea, and the phyllosphere, but again, imaging analyses would be needed to reveal if these sequences originated from bacteria associated with eukaryotes (Supplementary Information Table 18). “

What our analyses also showed is that the HMM profile we generated to identify IAPs was sufficient to detect these genes in diverse bacterial genomes, most likely because of the conserved RING and the BIR domains in IAPs.

*13. The statement in the abstract that IAPs were “repeatedly [acquired] through horizontal gene transfer (HGT) from their hosts in convergent acquisition” is not well-supported by the phylogenetic analysis. The tree does suggest repeated acquisition events, but rather than through independent events, the IAPs appear to have been acquired by the ancestor of the two *Endonucleobacter* species, followed by subsequent co-diversification and differential gene loss.*

We agree and thank the reviewer for this comment. Our revised IAP tree in Figure 4 has low bootstrap support at the ancestral nodes of the four "*Ca. Endonucleobacter*" clades. This means that we cannot discern if IAPs were acquired repeatedly in convergent evolution or not. We have now reworded the corresponding sentence in the abstract to – line 22 – 23:

“Comparative phylogenetic analyses revealed that “*Ca. Endonucleobacter*” acquired IAPs through horizontal gene transfer (HGT) from their hosts.”

*14. The observation of virus particles in *Endonucleobacter* cells is fascinating. However, as icosahedral capsids are not restricted to dsDNA viruses, the speculation that they might represent the host virus OsHV-1 seems far-fetched.*

We agree that this is purely speculative, and have revised our statement accordingly, both in the text and the legend of Extended Data Fig. 6 (revisions in bold):

Main text lines 405 – 410 :

"Support for this third explanation stems from our TEM analyses, which revealed **viral-like** structures inside “*Ca. E. childressi*” whose structure **resembles** double stranded DNA viruses, (**Extended Data Fig. 6**) (Davison et al., 2005). OsHV-1 is a double-stranded DNA virus and reportedly infects the nuclei of the oyster *Crassostrea gigas* (Renault et al., 2000), **but without evidence that the viral-like structures we observed in “*Ca. E. childressi*” are from OsHV-1, this explanation is purely speculative.**"

Legend of Extended Data Fig. 6 – line 1123 - 1126:

As we wrote in our manuscript (lines x-x), we did find OsHV-1 sequences (17% of the genome) in a library from a single *Bathymodiolus* individual. However, we obviously cannot show that these sequences originated from the virus-like particles we observed in our TEM images.

Is there any evidence for phages or pro-phages in the sequence assemblies?

22We used DeepVirFinder to screen our datasets but did not identify any phages in either dataset.

15. Data availability and traceability of genome annotation: The Endonucleobacter genome sequences at Genbank/ENA/DDBJ should include the complete annotation used in the present study. Supplementary tables should include the final locus tags used for submission of the annotated genome sequence.

Similarly, the G. childressi de novo transcriptome including the annotation must be submitted and available. The gene names and generic locus tags (trinity something) used in the manuscript for host genes are largely useless to readers interested in specific gene/protein sequences.

Are all original sequence data sets available in SRA?

Yes, we have deposited the raw reads and the bacterial MAGs to NCBI, under the following accession numbers: [PRJNA979916](https://www.ncbi.nlm.nih.gov/submitter/study/PRJNA979916).

We provide the host transcriptomes and the annotated genomes as used in this study under the following Zenodo link: <https://doi.org/10.5281/zenodo.11086256>

Currently we prefer not to deposit the host transcriptome as a reference transcriptome for *G. childressi*, as it would need further polishing to ensure a high enough quality for a reference transcriptome in one of the public databases. By making it available in Zenodo, we have, however, made our sequence data publicly accessible and our annotations traceable.

Minor points:

16. Figure 1a is not referred to in the manuscript; should be included in the first paragraph of the results and discussion section.

Corrected.

17. Figure 1a: Which Endonucleobacter species is shown? E. puteoserpentis?

Yes, the figure legend has been updated.

18. Figure 2: Gene expression data from which infection stage? From the “bulk transcriptomics” experiment?

We now clarify in the legend of Fig. 2 that it shows our analyses of the bulk transcriptomes, which does not provide information about the different infection stages:

Lines 210-212 : "Physiological reconstruction from bulk tissue transcriptomes and proteomes, based on RAST annotation and Pathway Tools v13.0 (Karp et al., 2010)."

19. L. 139, 143, "these were expressed 139 in *Ca. E. childressi*." : Are the respective genes or proteins expressed? At which infection stage?

To make this clearer, we moved the citation of Supp Table 2 and 3 in the text, so that the data is now better linked to our statement. We also clarify in the legend of Figure 2, as explained in our preceding reply, that the gene expression shown is from bulk transcriptomes. The color coding of circled genes and the corresponding letter P next to the circle genes show the level of transcriptomic and proteomic expression respectively. We have revised the text in the figure legend to make this clearer:

Line 212 – 217: "'*Ca. E. childressi*' transcriptomic expression is shown in circled genes, as TPMs normalized to *RecA* TPMs, for levels from not detected (black) through low (blue) to high (yellow) (color legend on bottom left). The expression levels of proteins are shown as colored "P symbols" next to their corresponding gene, with yellow showing high abundance (first quartile), turquoise medium abundance (second quartile) and blue low abundance (third quartile).

20. Supplementary Tables 3 and 4 appear to be identical.

We checked, these are different tables that just look very similar.

21. Supplementary Table 2 and 5: *E. puteoserpentis* is named *E. bathymodioli* in this table.

22. Figure 4: *E. puteoserpentis* is named *E. bathymodioli* in this figure.

23. Figure 4 legend: Typo, IQTREE should read IQ-TREE

24. Supplementary notes 3 and 4 are not referred to in the main text.

Thank you for pointing out these typos and issues in the above Comments 21-24, they have now been corrected.

Reviewer #3 (Remarks to the Author):

The introduction gives a relevant background to the bacterial clade of Ca. Endonucleobacter, which infects the nucleus of mussel cells. Clear research questions are stated that are related

24to cellular and molecular process that could support the replication and survival of the intranuclear Ca. Endonucleobacter cells.

The Material and Methods are described in a manner to reproduce the work and are performed to high technical standards.

The Results and Discussion section initially describes the localisation of Ca. Endonucleobacter in two mussels species, which show an interesting exclusion pattern with other symbiotic bacteria, such as sulfur- and / or methane-oxidizing bacteria. While I don't expect that this study can fully explain these exclusion patterns, I wonder if this could be picked up later in the Discussion or Conclusion given that sulfur- and / or methane-oxidizing bacteria would provide much energy to mussel cells, something that the immense reproduction of Ca. Endonucleobacter would certainly benefit from.

We agree that these exclusion patterns cannot be fully explained in our manuscripts, but have added the following in lines 98 -103:

“One explanation for this exclusion pattern could be that the apical surfaces of symbiont-containing bacteriocytes differ from those of other epithelial cells in bathymodoline mussels. Their bacteriocytes lack both the cilia and microvilli typical for epithelial cell surfaces (Frank et al. 2021). Epithelial surface structures are often targeted by pathogens for entering eukaryotic cells (), and their absence could hinder “Ca. Endonucleobacter” from infecting cells with symbionts.”

Genome reconstruction further defined that the Ca. Endonucleobacter in two mussels species represent two distinct species, which have much smaller genomes compared to related Endozoicomonas species. A loss of amino acid synthesis pathways is mentioned and it would have been of interest to understand what other genetic feature underpin the apparent genome reduction.

We agree with the reviewer that these questions are the next step in understanding this clade ranging from mutualist to parasites, however this was beyond the scope of the current study.

A careful and state-of-the-art genomic and transcriptomic analysis indicate that Ca. Endonucleobacter mostly utilises sugar, peptides and lipids, and likely not much nucleic acids. This is also consistent with the morphological and expression phenotypes observed for the host. I note though that the transcriptomic and proteomic data in Table S3, S4 and S7, which the section about gene expression refers to (i.e. L121ff), only shows 30-50 genes/proteins, which I assume is not all that is being detected. I could not find any headings or explanations

for the Supplementary Tables, which made understanding them quite challenging. Please provide more context for them.

We agree and now provide more information in the methods explaining which supplementary table represents which dataset.

The Ca. Endonucleobacter genomes further contain a number of bona fide IAPs and their expression patterns indicate a key role in facilitating the survival of the bacterium by blocking the apoptosis of the host. Evidence is presented that IAPs are acquired via multiple HGT as the authors identified eight clades where bacterial and non-bacterial sequences are interspersed. And finally, evidence is provided that Ca. Endonucleobacter is infected by a virus that could mediate such a HGT. The scenario described in the manuscript about how IAPs function and evolved are not overly speculative and very interesting.

The conclusion provides interesting connections to other work on intra-nuclear/cellular symbionts, Endozoicomonadaceae associations and eukaryote-to-prokaryote HGT, and this caps off a well-written and well-executed study that makes a significant contribution to our understanding of bacteria-host interactions.

We thank the reviewer for their kind comments.

Minor comments:

Line 19: Replace “digesting” by “digested”.

This sentence was revised to make this statement clearer to – line 18 – 19 :

"Instead, “Ca. Endonucleobacter” upregulated genes for importing and digesting sugars, lipids, amino acids and possibly mucin from its host."

Line 40: Replace “ubiquotous” with “ubiquitous”.

Line 93: Provide cross-link to Extended Data Fig. 1 here.

Line 184: Define “T3SS”.

Line 494: Fix “))”

L679: Please explain what you mean by “pseudoalign”.

The typos and issues above were all corrected and addressed.

L162: If the three Hahella genomes were used to root the tree, why is there then another root on the left side of the tree? This would indicate that some other sequence(s) was used to place the root between the two major clusters.

Thank you for pointing this out, we have now revised the tree shown in Figure 1.

Decision Letter, first revision:

Message: Our ref: NMICROBIOL-24020452A

26th July 2024

Dear Dr. Dubilier,

Thank you for your patience as we've prepared the guidelines for final submission of your Nature Microbiology manuscript, "An intranuclear bacterial parasite of deep-sea mussels expresses apoptosis inhibitors acquired from its host" (NMICROBIOL-24020452A). Please carefully follow the step-by-step instructions provided in the attached file, and add a response in each row of the table to indicate the changes that you have made. Please also check and comment on any additional marked-up edits we have proposed within the text. Ensuring that each point is addressed will help to ensure that your revised manuscript can be swiftly handed over to our production team.

In recognition of the time and expertise our reviewers provide to Nature Microbiology's editorial process, we would like to formally acknowledge their contribution to the external peer review of your manuscript entitled "An intranuclear bacterial parasite of deep-sea mussels expresses apoptosis inhibitors acquired from its host". For those reviewers who give their assent, we will be publishing their names alongside the published article.

Nature Microbiology offers a Transparent Peer Review option for new original research manuscripts submitted after December 1st, 2019. As part of this initiative, we encourage our authors to support increased transparency into the peer review process by agreeing to have the reviewer comments, author rebuttal letters, and editorial decision letters published as a Supplementary item. When you submit your final files please clearly state in your cover letter whether or not you would like to participate in this initiative. Please note that failure to state your preference will result in delays in accepting your manuscript for publication.

Cover suggestions

27COVER ARTWORK: We welcome submissions of artwork for consideration for our cover. For more information, please see our guide for cover artwork.

Nature Microbiology has now transitioned to a unified Rights Collection system which will allow our Author Services team to quickly and easily collect the rights and permissions required to publish your work. Approximately 10 days after your paper is formally accepted, you will receive an email in providing you with a link to complete the grant of rights. If your paper is eligible for Open Access, our Author Services team will also be in touch regarding any additional information that may be required to arrange payment for your article.

Please note that *Nature Microbiology* is a Transformative Journal (TJ). Authors may publish their research with us through the traditional subscription access route or make their paper immediately open access through payment of an article-processing charge (APC). Authors will not be required to make a final decision about access to their article until it has been accepted. Find out more about Transformative Journals

Reviewer #1:

Remarks to the Author:

The authors have addressed all of our major suggestions. Here are just a few tiny tweaks that might be easily implemented during the final revision

Line 18, 140-143, and elsewhere, "upregulated": in some places there seems to be something odd with the tense of some verbs

Line 53: please either specify which "amoeba" or, preferably, say "amoebas" or "amoebae" if you mean several different kinds of amoebae. Or better yet, just say protists since amoebae are protists.

Line 56: note that one exception that is somewhat similar to the one presented here comes from bacteria in the Holosporaceae family, that can replicate extensively in the nucleus of the host, Paramecium, and make it swell and burst

Line 61: The authors have fleshed out their comparisons with other systems with reviews and a few papers, but some more notes on primary literature might still be beneficial. For example, Trichorickettsia is a nuclear bacterium that does seem to occasionally eat chromatin (as opposed to Endonucleobacter) and has virus-like capsid particles like those discussed here (e.g.

<https://journals.plos.org/plosone/article?id=10.1371/journal.pone.0087718>)

Line 462, The discussion of HGT is improved, but the new section has one misrepresentation that should be deleted. The new section ends with the statement that bacteria get eukaryotic genes due to "strong selection for the acquisition of eukaryotic genes by prokaryotes (Keeling, 2024)", but the cited paper says nothing about that - it is about why eukaryotes don't get many bacterial genes. Indeed if anything this paper would argue against this statement since it mostly is saying close proximity does not matter. I don't think anyone says bacteria are under selection to get eukaryotic genes, and the statement makes no sense as the authors just finished (correctly) saying bacteria rarely get eukaryotic genes. So just delete the end of this sentence - your case is interesting and rare, and leave it at that.

Reviewer #4:
Remarks to the Author:

I have previously reviewed the manuscript and have added my second-round comments interleaved below.

1st round comment:

The introduction gives a relevant background to the bacterial clade of Ca. Endonucleobacter, which infects the nucleus of mussel cells. Clear research questions are stated that are related to cellular and molecular process that could support the replication and survival of the intranuclear Ca. Endonucleobacter cells.

The Material and Methods are described in a manner to reproduce the work and are performed to high technical standards.

The Results and Discussion section initially describes the localisation of Ca. Endonucleobacter in two mussels species, which show an interesting exclusion pattern with other symbiotic bacteria, such as sulfur- and / or methane-oxidizing bacteria. While I don't expect that this study can fully explain these exclusion patterns, I wonder if this could be picked up later in the Discussion or Conclusion given that sulfur- and / or methane-oxidizing bacteria would provide much energy to mussel cells, something that the immense reproduction of Ca. Endonucleobacter would certainly benefit from.

29We agree that these exclusion patterns cannot be fully explained in our manuscripts, but have added the following in lines 98 -103:

“One explanation for this exclusion pattern could be that the apical surfaces of symbiont-containing bacteriocytes differ from those of other epithelial cells in bathymodoline mussels. Their bacteriocytes lack both the cilia and microvilli typical for epithelial cell surfaces (Frank et al. 2021). Epithelial surface structures are often targeted by pathogens for entering eukaryotic cells (), and their absence could hinder “Ca. Endonucleobacter” from infecting cells with symbionts.”

2nd round comment:

Thanks.

1st round comment:

Genome reconstruction further defined that the Ca. Endonucleobacter in two mussels species represent two distinct species, which have much smaller genomes compared to related Endozoicomonas species. A loss of amino acid synthesis pathways is mentioned and it would have been of interest to understand what other genetic feature underpin the apparent genome reduction.

We agree with the reviewer that these questions are the next step in understanding this clade ranging from mutualist to parasites, however this was beyond the scope of the current study.

2nd round comment:

While I appreciate that the manuscript does not have to go into an extensive description of gene/function losses, I wonder why only amino acid synthesis is being mentioned (line 166). As in the subsequent section entitled “Ca. Endonucleobacter” gains its nutrition from host sugars, lipids and amino acids” you argue that the symbiont transports sugars and lipids (in addition to amino acid) into its cytoplasm, it would make sense to have at least some brief

statement if genome reduction also impacted anabolic or catabolic capacities for these compound classes. Alternatively, you could also consider deleting the statement about the loss of amino acid synthesis pathways in order to not give the wrong impression that this is the only functional group impacted by genome reduction.

1st round comment:

*A careful and state-of-the-art genomic and transcriptomic analysis indicate that *Ca. Endonucleobacter* mostly utilises sugar, peptides and lipids, and likely not much nucleic acids. This is also consistent with the morphological and expression phenotypes observed for the host. I note though that the transcriptomic and proteomic data in Table S3, S4 and S7, which the section about gene expression refers to (i.e. L121ff), only shows 30-50 genes/proteins, which I assume is not all that is being detected. I could not find any headings or explanations for the Supplementary Tables, which made understanding them quite challenging. Please provide more context for them.*

We agree and now provide more information in the methods explaining which supplementary table represents which dataset.

2nd round comment:

Thanks for the additional information, but it has still not provided much more clarity. Specifically, you now explain in Line 863ff that Table S4 is “A subset of the detected proteins [that] were used to supplement the metabolic model of “*Ca. E. childressi*” (Fig. 2, Supplementary Information Table 4)”. Does this mean that Table S4 only shows the subset that was used for the model or does Table S4 show all proteins that were detected. If the latter, then it is perhaps a bit of a concern that only 11 proteins were identified/detected. Similarly, Table S3 is now defined in line 1025ff as containing the RecA-normalised bulk gene expression level of “*Ca. E. childressi*”, but again the table shows only the expression of a relatively small number (48) of genes. I suspect these are not all genes whose expression was detected and were normalized by RecA, but as there are no clear table headings/legends (or column explanations), this is all unnecessarily hard to follow. There was no further information provided for Table S7. I hope you can provide further headings, legends and/or explanations (e.g. in the Supplementary Information file) that would make the supplementary data more accessible.

1st round comment:

*The *Ca. Endonucleobacter* genomes further contain a number of bona fide IAPs and their expression patterns indicate a key role in facilitating the survival of the bacterium by blocking the apoptosis of the host. Evidence is presented that IAPs are acquired via multiple HGT as the authors identified eight clades where bacterial and non-bacterial sequences are*

*interspersed. And finally, evidence is provided that *Ca. Endonucleobacter* is infected by a virus that could mediate such a HGT. The scenario described in the manuscript about how IAPs function and evolved are not overly speculative and very interesting.*

The conclusion provides interesting connections to other work on intra-nuclear/cellular symbionts, Endozoicomonadaceae associations and eukaryote-to-prokaryote HGT, and this caps off a well-written and well-executed study that makes a significant contribution to our understanding of bacteria-host interactions.

We thank the reviewer for their kind comments.

2nd round comment:

You are welcome.

1st round comment:

Minor comments:

Line 19: Replace “digesting” by “digested”.

This sentence was revised to make this statement clearer to – line 18 – 19 :

"Instead, “*Ca. Endonucleobacter*” upregulated genes for importing and digesting sugars, lipids, amino acids and possibly mucin from its host."

1st round comment:

Line 40: Replace “ubiquotous” with “ubiquitous”.

Line 93: Provide cross-link to Extended Data Fig. 1 here. Line

184: Define “T3SS”.

Line 494: Fix “))”

L679: Please explain what you mean by “pseudoalign”.

The typos and issues above were all corrected and addressed.

1st round comment:

*L162: If the three *Hahella* genomes were used to root the tree, why is there then another root on the left side of the tree? This would indicate that some other sequence(s) was used to place the root between the two major clusters.*

Thank you for pointing this out, we have now revised the tree shown in Figure 1.

2nd round comment:

Thanks for making all these changes.

Response to reviewers

Reviewer #1

The authors have addressed all of our major suggestions. Here are just a few tiny tweaks that might be easily implemented during the final revision

- 1.) *Line 18, 140-143, and elsewhere, “upregulated”*: in some places there seems to be something odd with the tense of some verbs.

We have reviewed the use of the work “upregulated” throughout the manuscript, but could not find any issues.

- 2.) *Line 53: please either specify which “amoeba” or, preferably, say “amoebas” or “amoebae” if you mean several different kinds of amoebae. Or better yet, just say protists since amoebae are protists.*

We have adjusted the text, and it now reads: “Intranuclear bacteria have rarely been described in animals but are well known from protists.”

- 3.) *Line 56: note that one exception that is somewhat similar to the one presented here comes from bacteria in the Holosporaceae family, that can replicate extensively in the nucleus of the host, Paramecium, and make it swell and burst*

- 4.) *Line 61: The authors have fleshed out their comparisons with other systems with reviews and a few papers, but some more notes on primary literature might still be beneficial. For example, Trichorickettsia is a nuclear bacterium that does seem to occasionally eat chromatin (as opposed to Endonucleobacter) and has virus-like capsid particles like those discussed here (e.g.*

<https://journals.plos.org/plosone/article?id=10.1371/journal.pone.0087718>)

We agree with the reviewers that these are fascinating observations. Unfortunately, Nature Microbiology guidelines required that we remove a substantial number of citations, and therefore could not add additional citations.

2

5.) *Line 462, The discussion of HGT is improved, but the new section has one misrepresentation that should be deleted. The new section ends with the statement that bacteria get eukaryotic genes due to “strong selection for the acquisition of eukaryotic genes by prokaryotes (Keeling, 2024)”, but the cited paper says nothing about that - it is about why eukaryotes don’t get many bacterial genes. Indeed if anything this paper would argue against this statement since it mostly is saying close proximity does not matter. I don’t think anyone says bacteria are under selection to get eukaryotic genes, and the statement makes no sense as the authors just finished (correctly) saying bacteria rarely get eukaryotic genes. So just delete the end of this sentence - your case is interesting and rare, and leave it at that.*

We have reworded this section and it now reads:

HGT from eukaryotes to prokaryotes is assumed to be disfavored for several reasons including the presence of eukaryotic introns as barriers to genetic transfer of genes to prokaryotes, and the lack of eukaryotic metabolic versatility compared to bacteria⁵². Eukaryotes have numerous genes and pathways for interacting with both beneficial and parasitic bacteria.

Reviewer #4

I have previously reviewed the manuscript and have added my second-round comments interleaved below.

1.) *1st round comment:*

*The introduction gives a relevant background to the bacterial clade of *Ca. Endonucleobacter*, which infects the nucleus of mussel cells. Clear research questions are stated that are related to cellular and molecular process that could support the replication and survival of the intra-nuclear *Ca. Endonucleobacter* cells. The Material and Methods are described in a manner to reproduce the work and are performed to high technical standards. The Results and Discussion section initially describes the localisation of *Ca. Endonucleobacter* in two mussels species, which show an interesting exclusion pattern with other symbiotic bacteria, such as sulfur- and / or methane-oxidizing bacteria. While I don’t expect that this study can fully explain these exclusion patterns, I wonder if this could be picked up later in the Discussion or Conclusion given that sulfur- and / or methane-oxidizing bacteria would provide much energy to mussel cells, something that*

3the immense reproduction of Ca. Endonucleobacter would certainly benefit from.

We agree that these exclusion patterns cannot be fully explained in our manuscripts, but have added the following in lines 98 -103:

“One explanation for this exclusion pattern could be that the apical surfaces of symbiont-containing bacteriocytes differ from those of other epithelial cells in bathymodoline mussels. Their bacteriocytes lack both the cilia and microvilli typical for epithelial cell surfaces (Frank et al. 2021). Epithelial surface structures are often targeted by pathogens for entering eukaryotic cells (), and their absence could hinder “Ca. Endonucleobacter” from infecting cells with symbionts.”

2nd round comment:

Thanks.

2.) *1st round comment:*

Genome reconstruction further defined that the Ca. Endonucleobacter in two mussels species represent two distinct species, which have much smaller genomes compared to related Endozoicomonas species. A loss of amino acid synthesis pathways is mentioned and it would have been of interest to understand what other genetic feature underpin the apparent genome reduction.

We agree with the reviewer that these questions are the next step in understanding this clade ranging from mutualist to parasites, however this was beyond the scope of the current study.

2nd round comment:

While I appreciate that the manuscript does not have to go into an extensive description of gene/function losses, I wonder why only amino acid synthesis is being mentioned (line 166). As in the subsequent section entitled “Ca. Endonucleobacter” gains its nutrition from host sugars, lipids and amino acids” you argue that the symbiont transports sugars and lipids (in addition to amino acid) into its cytoplasm, it would make sense to have at least some brief statement if genome reduction also impacted anabolic or catabolic capacities for these compound classes. Alternatively, you could also consider deleting the statement about the loss of amino acid synthesis pathways in order to not give the wrong impression that this is the only functional group impacted by genome reduction.

We have adjusted the text and qualify our statement as follows:

“Ca. Endonucleobacter” genomes were smaller, had reduced GC contents, and encoded considerably less amino acid synthesis pathways than *Endozoicomonas* species (Fig. 1b). A detailed, comprehensive analysis of genome reduction in “Ca. Endonucleobacter” is planned in a future study to predict its impact on metabolic pathways in these intranuclear pathogens.

3.) 1st round comment:

*A careful and state-of-the-art genomic and transcriptomic analysis indicate that *Ca. Endonucleobacter* mostly utilises sugar, peptides and lipids, and likely not much nucleic acids. This is also consistent with the morphological and expression phenotypes observed for the host. I note though that the transcriptomic and proteomic data in Table S3, S4 and S7, which the section about gene expression refers to (i.e. L121ff), only shows 30-50 genes/proteins, which I assume is not all that is being detected. I could not find any headings or explanations for the Supplementary Tables, which made understanding them quite challenging. Please provide more context for them.*

We agree and now provide more information in the methods explaining which supplementary table represents which dataset.

2nd round comment:

*Thanks for the additional information, but it has still not provided much more clarity. Specifically, you now explain in Line 863ff that Table S4 is “A subset of the detected proteins [that] were used to supplement the metabolic model of “*Ca. E. childressi*” (Fig. 2, Supplementary Information Table 4)”. Does this mean that Table S4 only shows the subset that was used for the model or does Table S4 show all proteins that were detected. If the latter, then it is perhaps a bit of a concern that only 11 proteins were identified/detected. Similarly, Table S3 is now defined in line 1025ff as containing the *RecA*-normalised bulk gene expression level of “*Ca. E. childressi*”, but again the table shows only the expression of a relatively small number (48) of genes. I suspect these are not all genes whose expression was detected and were normalized by *RecA*, but as there are no clear table headings/legends (or column explanations), this is all unnecessarily hard to follow. There was no further information provided for Table S7. I hope you can provide further headings, legends and/or explanations (e.g. in the Supplementary Information file) that would make the supplementary data more accessible.*

We have added comprehensive table legends for all 17 Supplementary tables. For example, the reviewer is correct that Supplementary Table 4 is a shortlist of proteins detected by the overall analysis, and we now clarify this for each Supplementary table where this is the case. For example, for Supp Table 4, the header reads now: “Subset of “*Ca. E. childressi*” factors extracted from the global bulk proteomic analysis (Supplementary information table 10) to inform the metabolic model (Fig. 2).” followed by the explanation of the individual columns.

4.) 1st round comment:

*The *Ca. Endonucleobacter* genomes further contain a number of bona fide IAPs and*

5

their expression patterns indicate a key role in facilitating the survival of the bacterium by blocking the apoptosis of the host. Evidence is presented that IAPs are acquired via multiple HGT as the authors identified eight clades where bacterial and non-bacterial sequences are interspersed. And finally, evidence is provided that Ca. Endonucleobacter is infected by a virus that could mediate such a HGT. The scenario described in the manuscript about how IAPs function and evolved are not overly speculative and very interesting.

The conclusion provides interesting connections to other work on intra-nuclear/cellular symbionts, Endozoicomonadaceae associations and eukaryote-to-prokaryote HGT, and this caps off a well-written and well-executed study that makes a significant contribution to our understanding of bacteria-host interactions.

We thank the reviewer for their kind comments.

2nd round comment:

You are welcome.

5.) *1st round comment:*

Minor comments:

Line 19: Replace “digesting” by “digested”.

This sentence was revised to make this statement clearer to – line 18 – 19 :

"Instead, “Ca. Endonucleobacter” upregulated genes for importing and digesting sugars, lipids, amino acids and possibly mucin from its host."

1st round comment:

Line 40: Replace “ubiquotous” with “ubiquitous”.

Line 93: Provide cross-link to Extended Data Fig. 1 here.

Line 184: Define “T3SS”.

Line 494: Fix “)”

L679: Please explain what you mean by “pseudoalign”.

The typos and issues above were all corrected and addressed.

1st round comment:

L162: If the three Hahella genomes were used to root the tree, why is there then another root on the left side of the tree? This would indicate that some other sequence(s) was used to place the root between the two major clusters.

Thank you for pointing this out, we have now revised the tree shown in Figure 1.

2nd round comment:

Thanks for making all these changes.

Final Decision Letter:

Message: 13th August 2024

Dear Nicole,

I am pleased to accept your Article "An intranuclear bacterial parasite of deep-sea mussels expresses apoptosis inhibitors acquired from its host" for publication in Nature Microbiology. Thank you for having chosen to submit your work to us and many congratulations.

7Please note that *Nature Microbiology* is a Transformative Journal (TJ). Authors may publish their research with us through the traditional subscription access route or make their paper immediately open access through payment of an article-processing charge (APC). Authors will not be required to make a final decision about access to their article until it has been accepted. Find out more about Transformative Journals

With kind regards,